# A catalytically active oscillator made from small organic molecules

Matthijs ter Harmsel[1], Oliver R. Maguire[2], Sofiya A. Runikhina[1], Albert S. Y. Wong[3], Wilhelm T. S. Huck[2✉] & Syuzanna R. Harutyunyan[1✉]

Oscillatory systems regulate many biological processes, including key cellular functions such as metabolism and cell division, as well as larger-scale processes such as circadian rhythm and heartbeat[1–4]. Abiotic chemical oscillations, discovered originally in inorganic systems[5,6], inspired the development of various synthetic oscillators for application as autonomous time-keeping systems in analytical chemistry, materials chemistry and the biomedical field[7–17]. Expanding their role beyond that of a pacemaker by having synthetic chemical oscillators periodically drive a secondary function would turn them into significantly more powerful tools. However, this is not trivial because the participation of components of the oscillator in the secondary function might jeopardize its time-keeping ability. We now report a small molecule oscillator that can catalyse an independent chemical reaction in situ without impairing its oscillating properties. In a flow system, the concentration of the catalytically active product of the oscillator shows sustained oscillations and the catalysed reaction is accelerated only during concentration peaks. Augmentation of synthetic oscillators with periodic catalytic action allows the construction of complex systems that, in the future, may benefit applications in automated synthesis, systems and polymerization chemistry and periodic drug delivery.

Nearly all known synthetic oscillatory reactions are based on redox chemistry[7,9,12], apart from a few examples based on biomolecules[18–20], small organic molecules[21,22] or supramolecular assemblies[23,24]. Of these oscillators, only the Belousov–Zhabotinsky reaction has been used to control another reaction, but its product was severely affected by the components of the oscillator[25]. The operational simplicity of organocatalytic reactions, the robustness and non-toxicity of the catalysts, their broad functional group tolerance and the diversity of organocatalysts[26,27] offer strong potential[28] for the modular design and application of organocatalytic oscillators. Moreover, such oscillators would benefit from readily available building blocks, the ability to function under non-aggressive conditions and easy modification via the power of organic chemistry.

Our catalytically active oscillator is made from small organic molecules and is based on the principles of aminocatalysis. It makes use of autocatalytic 9-fluorenylmethoxycarbonyl (Fmoc) group deprotection[29,30] and acetylation of amines. Using an open flow system, sustained oscillations are realized over a range of different conditions. The key component of the oscillator is an organocatalyst that promotes chemical reactions via enamine and/or base catalysis. The reactants that form the oscillator and those involved in the catalysed reaction are mixed in the same reactor and form a system in which each component serves a unique purpose with minimal interference between oscillation and catalysis.

The design of our system follows an established principle for chemical oscillators: a combination of a positive and negative feedback loop(s) leads to alternating growth and decay phases, giving rise to oscillations[6,31,32]. We chose to base our system on autocatalytic Fmoc deprotection (Fig. 1). Fmoc is a base labile-protecting group that can be used to protect amines. We selected piperidine (**1**) as the amine because it is (1) a commonly used organocatalyst[26] and (2) sufficiently basic to catalyse the deprotection of Fmoc groups. This makes the deprotection of Fmoc-piperidine (**2**) autocatalytic[29,30], thereby providing the positive-feedback loop. The negative-feedback loop is delivered by two inhibition reactions, one fast and one slow, that prevent piperidine from reacting as a base, counteracting the autocatalytic pathway. Acetylation with phenyl acetates is used for this purpose, transforming piperidine into an amide. The rate of acetylation can be readily tuned by changing the substituents on the phenyl ring. We chose *p*-nitrophenyl acetate (**3**) as the fast inhibitor that suppresses the autocatalytic reaction, creating a lag phase until it is consumed and autocatalytic growth of **1** takes over. Phenyl acetate (PhOAc, **4**) acts as the slow inhibitor that returns the concentration of piperidine to the initial state after autocatalysis has finished. Finally, the deprotection of **2** by a base other than **1** serves as a trigger reaction that continuously creates the autocatalyst at a low rate, thereby kick-starting the autocatalytic reaction. This deprotection has to be orthogonal to the inhibition reactions, leading us to choose the simplest tertiary analogue of piperidine, *N*-methylpiperidine (**5**), as our trigger reagent.

First, we set out to investigate whether our design could produce a pulse—a single cycle of exponential growth followed by decay back to baseline—in batch. After initial optimization (Supplementary Fig. 2) we

[1]Stratingh Institute for Chemistry, University of Groningen, Groningen, the Netherlands. [2]Institute for Molecules and Materials, Radboud University, Nijmegen, the Netherlands. [3]Department of Molecules and Materials, University of Twente, Enschede, the Netherlands. ✉e-mail: w.huck@science.ru.nl; s.harutyunyan@rug.nl

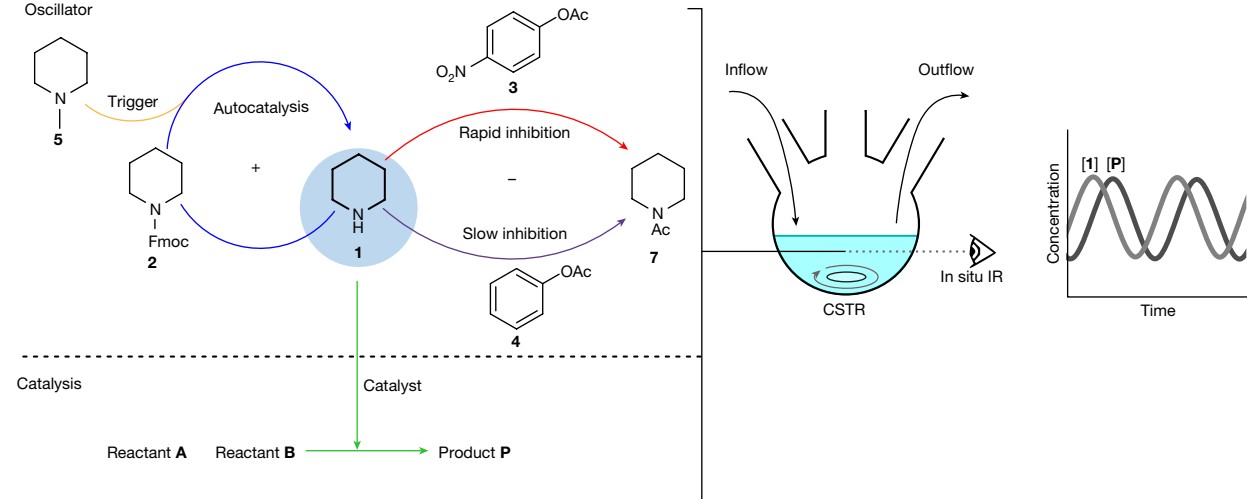

**Fig. 1 | Design of the organocatalytic oscillator.** The chemical reaction network comprises five reactions: autocatalytic Fmoc-piperidine (**2**) deprotection (blue), *N*-methylpiperidine (**5**)-catalysed Fmoc-piperidine (**2**) deprotection (orange), fast inhibition via acetylation by *p*-nitrophenyl acetate (**3**, red), slow inhibition by phenyl acetate (**4**, purple)—which irreversibly removes piperidine (**1**) from the reaction network as *N*-acetyl piperidine (**7**)—and the piperidine (**1**)-catalysed chemical reaction between reactants **A** and **B** yielding product **P**, green. All these reactions take place in the same reaction vessel—a CSTR with continuous inflow of reagents **2**–**5** and outflow of reaction mixture. The oscillating concentrations in the reaction vessel are monitored with in situ infrared spectroscopy.

arrived at the following conditions: 100 mM Fmoc-piperidine (**2**), 5 mM *p*-nitrophenyl acetate (**3**), 1 M PhOAc (**4**) and 5 mM *N*-methylpiperidine (**5**) in DMSO-$d_6$ (Fig. 2a). Fmoc deprotection is a two-step process that starts with deprotonation of **2** to form dibenzofulvene (**6**, DBF) and a carbamic acid, followed by decarboxylation of the latter to form **1**. Carbamic acid persists in DMSO under atmospheric conditions[33] but at elevated temperature (60 °C) the decarboxylation reaction occurs

rapidly. Concentrations were monitored over time using a combination of in situ infrared (IR) spectroscopy and $^1$H-NMR spectroscopy (Supplementary Information), with 1,3,5-trimethoxybenzene as an internal standard (Fig. 2b).

A lag phase (Fig. 2b, phase I) is observed in the conversion of **2**, showing that the rate of acetylation by **3** is sufficiently high to suppress the autocatalytic loop. This is supported by the concentrations of *N*-acetyl

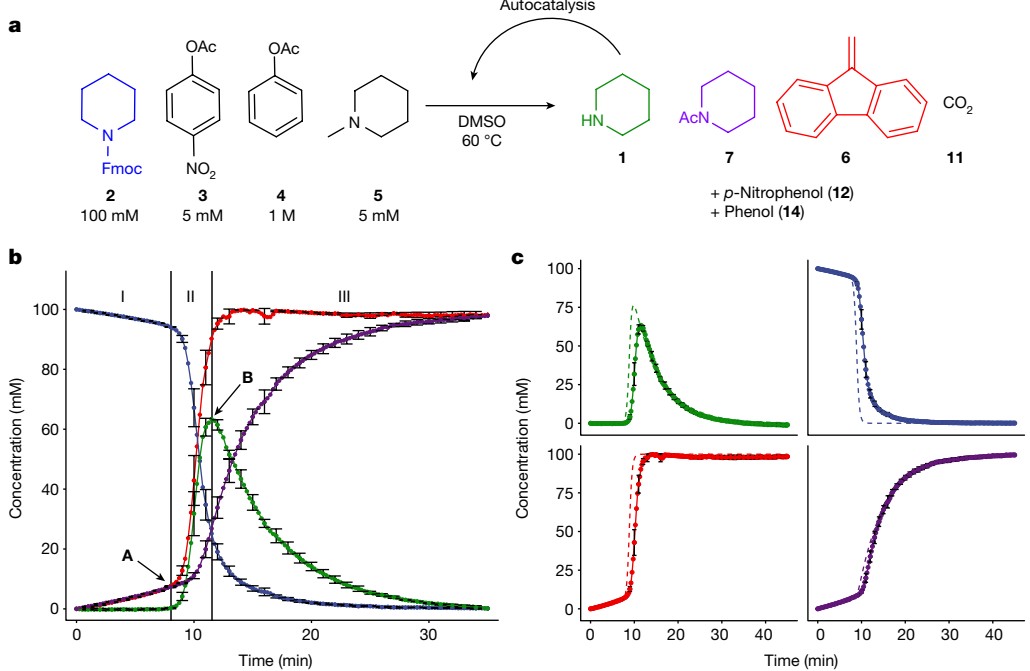

**Fig. 2 | Single-pulse experiment and comparison between experimental and modelled data. a**, Scheme showing the conditions used for single-pulse experiments. **b**, Pulse carried out at 60 °C using 100 mM Fmoc-piperidine (**2**), 5 mM *p*-nitrophenyl acetate (**3**), 1 M phenyl acetate (**4**) and 5 mM *N*-methylpiperidine (**5**) in DMSO. The reaction was followed by in situ infrared spectroscopy, with concentrations estimated using $^1$H-NMR spectroscopy (details in Supplementary Information): piperidine (**1**, green), Fmoc-piperidine (**2**, blue), DBF (**6**, red), PipAc (**7**, purple). The concentration of piperidine (**1**) was determined by taking the difference between the concentrations of DBF (**6**) and PipAc (**7**). Curves are an average of two experiments; for visual clarity, error bars (s.d.) are shown only for every 1 min. The pulse is divided into three distinct phases: a lag phase (I), exponential growth (II, starts at arrow A) and decay (III, starts at arrow B). **c**, The experimental data are compared with a model of the system based on ordinary differential equations (dashed lines). A version of the figure including all error bars is available in Supplementary Fig. 17.

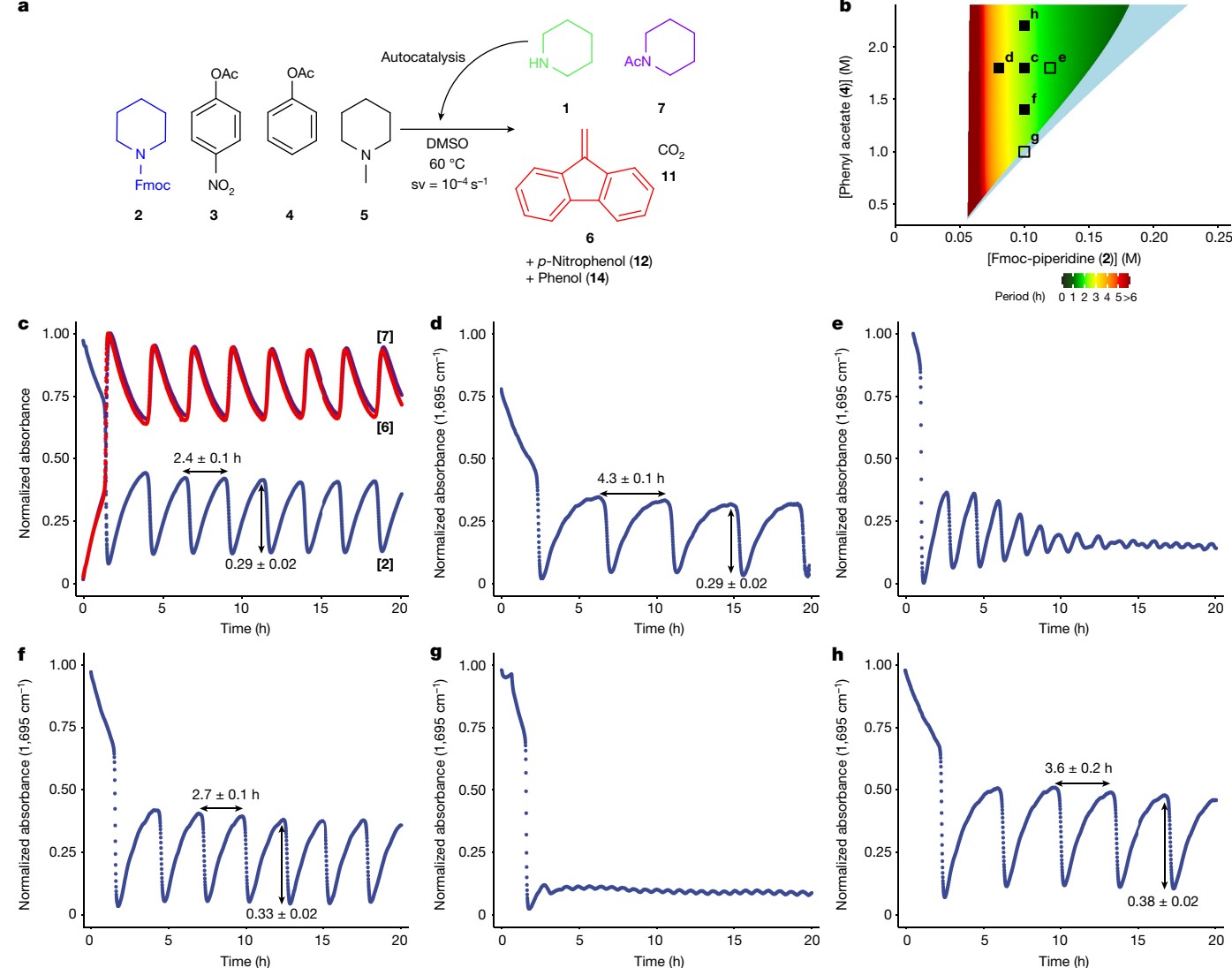

**Fig. 3 | Sustained oscillations under out-of-equilibrium conditions in flow using a CSTR. a**, Reaction scheme for flow oscillation experiments. **b**, Oscillation space predicted by the model: the region coloured by a gradient from red to green indicates the ranges of **4** and **2** that are predicted to support sustained oscillations with [**3**] = 30 mM, [**5**] = 5 mM and a space velocity of $10^{-4}$ s$^{-1}$ (defined as flow rate divided by CSTR volume); colour coding shows the predicted period (h). In the light blue region, dampened oscillations are predicted to take place and in the uncoloured region no oscillations occur. Filled squares indicate where experiments found sustained oscillations, and open squares where dampened oscillations were observed. **c**, Flow experiment carried out in CSTR at 60 °C using [**2**] = 100 mM, [**3**] = 30 mM, [**4**] = 1.8 M and [**5**] = 5 mM in DMSO with a space velocity of $10^{-4}$ s$^{-1}$. Oscillation was monitored using in situ infrared spectroscopy. Sustained oscillations were obtained for **2** (blue), **6** (red) and **7** (purple). Period and amplitude are mean plus s.d. from a single experiment. **d**–**h**, The oscillation space was investigated by carrying out experiments with the following deviations from the conditions shown in **c**: **d**, [**2**] = 80 mM, **e**, [**2**] = 120 mM, **f**, [**4**] = 1.4 M, **g**, [**4**] = 1.0 M, **h**, [**4**] = 2.2 M.

piperidine (PipAc, **7**), the inhibition reaction product, and **6**, the Fmoc remnant, which increase equally rapidly whereas the concentration of **1** remains negligible. Once all of fast inhibitor **3** has been consumed (Fig. 2b, phase II), the concentration of **2** plummets and that of **1** rises exponentially as the autocatalytic deprotection reaction accelerates. When most of **2** is consumed, the effect of the slow inhibition reaction of **4** with **1** starts to dominate (Fig. 2b, phase III), consuming **1** and completing the pulse in the piperidine concentration. A similar experiment was performed using 1,8-diazabicyclo[5.4.0]undec-7-ene (DBU) as an alternative trigger (Supplementary Fig. 3), whereas in another experiment the piperidine concentration was monitored directly via ultraviolet-visible spectroscopy with the pH indicator bromothymol blue (Supplementary Fig. 5).

Having obtained a pulse, we determined experimentally the rate laws for all reactions involved (Supplementary Table 8) and built a model of the system based on ordinary differential equations (Supplementary Information). In Fig. 2c the measured individual component concentrations are compared with model predictions, showing that the model represents the data well. Transforming a single pulse into a series of oscillations is possible under out-of-equilibrium conditions. This is achieved by placing the system in flow using a continuous stirred tank reactor (CSTR) in which fresh starting materials (**2**–**5**) are continuously added and reaction mixture is continuously removed. Such oscillations can either be sustained—constant amplitude—or dampened—amplitude decreases over time until a steady state is reached. The model of the pulse system was adapted to assess the effects of various experimental conditions, to predict which combinations of initial concentrations produce sustained oscillations under continuous flow (Fig. 3a,b). We use space velocity (sv) as the parameter to describe the rate of the inflow of the reactants and the outflow of the reaction mixture.

For the purposes of this screening we defined sustained oscillations as giving at least two pulses with no decrease of amplitude over time. The model predicts sustained oscillations for reactant concentrations of [**4**] = 0.4–2.0 M, [**2**] = 60–160 mM, [**3**] = 30 mM and [**5**] = mM, and a space velocity of $10^{-4}$ s$^{-1}$. The region of sustained oscillations is flanked on the high [**2**] and low [**4**] sides by a region of dampened oscillations (Fig. 3b). Complete results are available in Supplementary Data 1 and 2.

We started our oscillation experiments in flow with a set of conditions in the centre of the predicted sustained oscillation regime: [**2**] = 0.1 M and [**4**] = 1.8 M. In the CSTR, concentrations are monitored with in situ infrared spectroscopy; because absolute concentrations cannot be determined, normalized absorbance is reported (Supplementary Information). As predicted, the system produced sustained oscillations in the concentration of Fmoc-piperidine for 20 h with a period of 2.4 ± 0.1 h and a consistent amplitude of 0.29 ± 0.02. Antiphase oscillations were also observed in the concentrations of **6** and **7** (Fig. 3c). Notably, oscillations become sustained immediately after the first peak, indicating that the behaviour is robust and withstands the initial conditions[34].

To investigate the effect of the concentrations of core system components we chose to vary [**2**] and [**4**] (Fig. 3b) because these influence the strength of the positive- and negative-feedback loop, respectively. Decreasing [**2**] from 100 to 80 mM led to an increase in period to 4.3 ± 0.1 h (Fig. 3d) in accordance with the expected behaviour: the slower trigger reaction delays the start of the autocatalytic phase. Increasing [**2**] to 120 mM resulted in dampened oscillations (Fig. 3e). Lowering [**4**] from 1.8 to 1.4 M had only a minor effect on oscillations (Fig. 3f), but lowering concentration further to 1.0 M (Fig. 3g) gave a steady state after one pulse. Increasing [**4**] to 2.2 M (Fig. 3h) led to a significant increase in both period (3.6 ± 0.2 h) and amplitude (0.38 ± 0.02), contrary to the model's prediction that period would be constant following changes in [**4**] (Fig. 3b and Supplementary Fig. 18). This mismatch suggests that the model does not precisely capture all effects, such as the influence of solvent polarity, on reaction rates.

These experiments demonstrate not only the robustness of the oscillator—by showing that sustained oscillations can be obtained across a wide range of conditions—but also that the oscillation properties (period and amplitude) can be tuned by modifying the core component concentrations. Importantly, the experiments confirm that our model is able to predict which conditions yield sustained oscillations, although experimentally the oscillatory regime is slightly smaller: a higher concentration of **2** (Fig. 3e) shows dampened rather than the predicted sustained oscillations (Supplementary Fig. 18).

Having developed a new small-molecule oscillator with an organocatalyst at its core, we questioned whether catalysis can be coupled to the oscillator as a secondary function. For this purpose we chose piperidine-catalysed Knoevenagel condensation[35,36] between salicyl aldehyde (**8**) and dimethyl malonate (**9**) that forms 3-(methoxycarbonyl)coumarin (**10**) following intramolecular cyclization. This particular example of Knoevenagel reaction was selected because the resulting coumarin product derivatives are common scaffolds for biologically active compounds[37] and fluorophores[38]. Because a catalyst such as **1** is required for this reaction to proceed, the product coumarin **10** should be formed only when there is a sufficiently high concentration of **1** in solution, causing the concentration of **10** to oscillate in tandem with that of catalyst **1**.

We studied the system in a single-pulse experiment initially (Supplementary Fig. 22), whereupon two important observations were made. First, no major changes were seen in the growth–decay cycle of piperidine compared with the pulse in the absence of the catalytic reaction, apart from a slight broadening of the peak of [**1**], confirming that there is minimal interference between the catalysed reaction and oscillator system. Second, a sharp increase in the concentration of catalysed product **10** coincides with the peak in [**1**]. To demonstrate the robustness of this ability to couple catalysis to oscillations, we

studied other reactions as well. A catalytic pulse can also be achieved for another Knoevenagel reaction, and for a mechanistically distinct Claisen–Schmidt/Michael cascade (Supplementary Fig. 25).

To realize sustained oscillations in the catalytic synthesis of **10**, we used the same conditions as for the initial flow experiment but added 200 mM to each of **8** and **9** (Fig. 4c). Stable oscillations were observed in the absorption bands associated with **6**, **8** and **10**, with a period of 2.2 ± 0.2 h. This period is within the experimental error of that obtained without the presence of piperidine-catalysed Knoevenagel condensation, and demonstrates crucially that a load can be applied to our oscillator in a catalytic fashion without significantly altering the core characteristics of the oscillator itself.

To probe the stability of our catalytic oscillator we performed experiments at the same concentration combinations as carried out without coupled catalysis (Fig. 4b). The behaviour is comparable, with the exception of the experiment with a lower concentration of slow inhibitor **4** in which the oscillations are dampened rather than sustained. Thus, performing a catalytic reaction does not significantly change the characteristics of the oscillator, other than slightly contracting the oscillatory regime. We also investigated the effect of fast inhibitor **3** and found that decreasing its concentration to 0.025 M led to dampened oscillations; however, increasing it to 0.035 M yielded sustained oscillations with an increased period of 3.1 ± 0.1 h (Supplementary Fig. 27). Further experiments performed to examine the dependence on temperature showed that increasing the temperature to 70 °C led to dampened oscillations (Fig. 4h) but lowering it to 50 °C produced sustained oscillations with a significantly longer period (Fig. 4i). Finally, we carried out an experiment in which we perturbed the system after two pulses by raising the temperature from 60 to 70 °C for 30 min (Fig. 4j). This caused the exponential phase to occur sooner, thereby shortening the period of oscillation. However, once perturbation had ended oscillation quickly returned to its normal period (2.4 ± 0.2 h), demonstrating that our catalytic oscillator is robust against short-term temperature changes.

A catalytic oscillator such as the one developed in this work could potentially perform various functions in more extensive chemical reaction networks. For example, pulses of catalysis could be used to enhance selectivity in chemical transformations. A catalytic oscillator can act as a filter when applied to a mixture of compounds that undergo the same catalytic transformation, and would normally generate a mixture of products in the presence of a non-oscillating catalyst. In a conventional CSTR setup in which the catalyst is present continuously, the steady-state conversion reached by the less reactive substrate is determined by flow rate and reaction kinetics. With a catalytic oscillator, however, the catalyst is available for only short periods during which the less reactive substrate cannot reach a high conversion. For a simulation demonstrating this concept see Supplementary Fig. 29.

To explore this concept of achieving enhanced chemoselectivity, we performed a competition experiment for the condensation reaction with **9** (Fig. 5) between aldehyde **8** and *p*-hydroxybenzaldehyde (**15**). First, we confirmed that selectivity is obtained with a single pulse under batch conditions but not without the pulse (Supplementary Fig. 32). In a single pulse, **8** is converted fully whereas **15** reaches only 10% conversion and then remains constant because the drop in catalyst concentration halts the reaction. In the absence of a pulse, **8** is converted fully after which **15** continues to react, reaching 50% conversion after 60 min. Subsequently we performed competition experiments with the oscillator (Fig. 5a). Whereas **8** reaches a mean conversion of 56% with peaks of 97%, the conversion of **15** to **16** is negligible, averaging below 1% (Supplementary Table 13). To allow unbiased comparison with the non-oscillation case we determined the peak concentration of **1** during oscillations experimentally at 22 ± 1 mM (Supplementary Fig. 33). We then carried out a control experiment without oscillations by removing slow inhibitor **4** and tuning the steady state [**1**] to be the same as peak **1** concentration—that is, 22 mM (Fig. 5b). In the absence

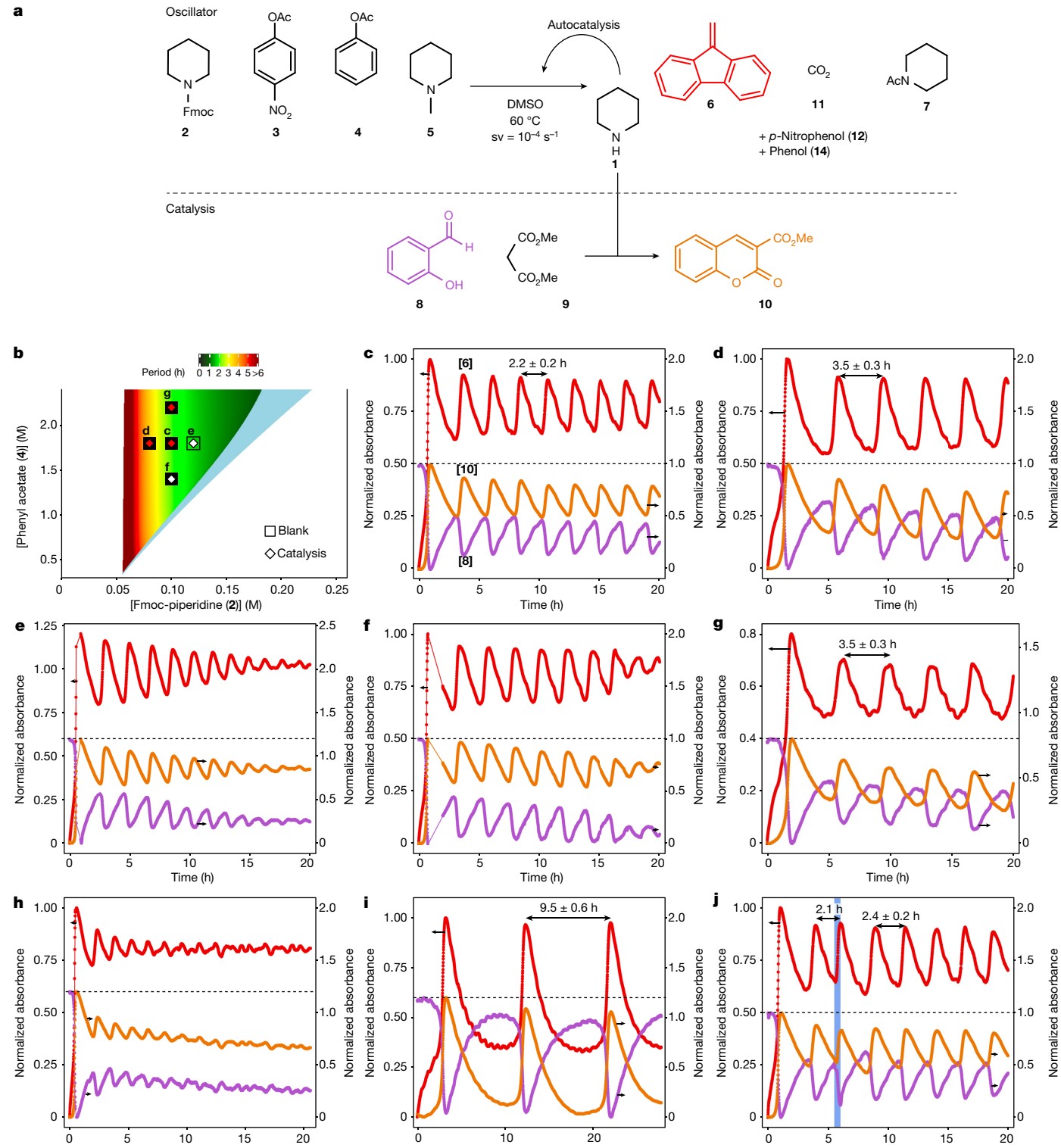

**Fig. 4 | Knoevenagel condensation controlled by an organic oscillator within the same CSTR. a**, Reaction scheme for a catalytic reaction coupled to flow oscillations. **b**, Oscillation space predicted by the oscillator model in the absence of catalysis: the region coloured in a gradient from red to green indicates the ranges of [**4**] and [**2**] predicted to support sustained oscillations with [**3**] = 30 mM, [**5**] = 5 mM and a space velocity of $10^{-4}$ s$^{-1}$; colour coding indicates the predicted period. In the light blue region, dampened oscillations are predicted to take place and in the uncoloured region no oscillations occur. Squares represent oscillator experiments carried out without catalysis, as in Fig. 3b; filled squares indicate where experiments found sustained oscillations and open squares dampened oscillations. Diamonds represent oscillator experiments carried out with

catalysis; red diamonds indicate where experiments found sustained oscillations and white diamonds dampened oscillations. **c**, Flow experiment carried out in CSTR at 60 °C using [**2**] = 100 mM, [**3**] = 30 mM, [**4**] = 1.8 M, [**5**] = 5 mM, [**8**] = 200 mM and [**9**] = 200 mM in DMSO, with a space velocity of $10^{-4}$ s$^{-1}$. Sustained oscillations are obtained for **6** (red), **8** (pink) and **10** (orange). The reported period is the mean with s.d. from a single experiment. **d**–**i**, The oscillation space was investigated by carrying out experiments with the following deviations from the conditions shown in **c**: **d**, [**2**] = 80 mM; **e**, [**2**] = 120 mM; **f**, [**4**] = 1.4 M; **g**, [**4**] = 2.2 M; **h**, 70 °C rather than 60 °C; **i**, 50 °C rather than 60 °C; **j**, flow experiment carried out in CSTR using the same method as in **c**, but after two pulses the system was perturbed by raising the temperature to 70 °C for 30 min (blue-shaded area).

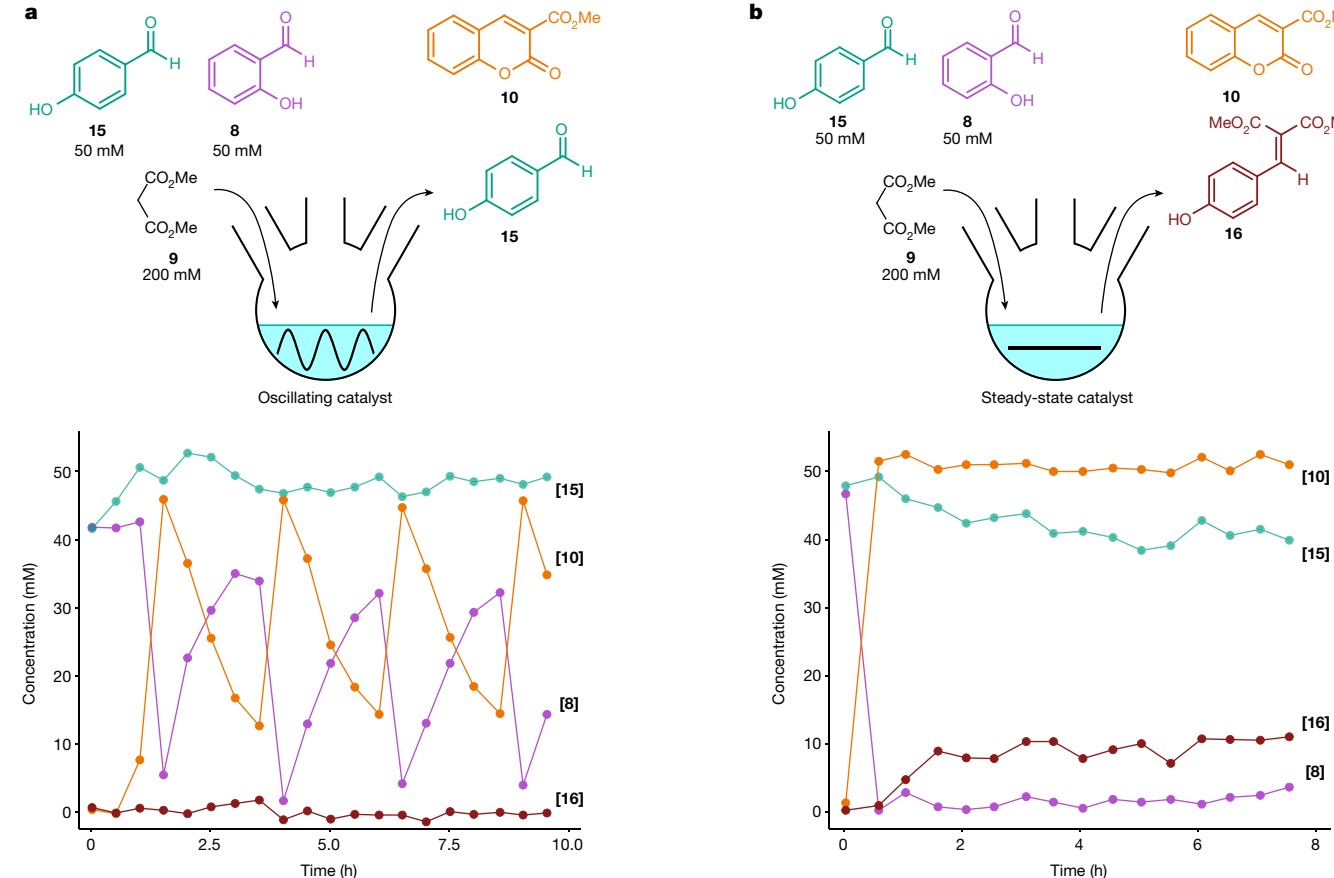

**Fig. 5 | Oscillation-enhanced selectivity. a,b**, Oscillation experiment (**a**) showing selectivity for salicyl aldehyde (**8**) over *p*-hydroxybenzaldehyde (**15**) contrasted with a control experiment in flow in the absence of oscillations (**b**). These competition experiments were carried out using conditions and starting ingredient concentrations that result in peak/steady-state concentrations of piperidine of 22 ± 1/19 ± 1 mM (mean plus s.d.; Supplementary Fig. 33). In the oscillating conditions (**a**) the concentrations of **8** (pink) and the associated product 3-(methoxycarbonyl)coumarin (**10**, orange) were found to oscillate, whereas the concentrations of **15** (teal) and its associated product dimethyl

(*p*-hydroxybenzylidene)malonate (**16**, brick red) remain nearly constant with less than 6% conversion of **15** to **16**. By contrast, in the non-oscillating control experiment with steady concentrations of piperidine (**b**) that equal the peak concentration in their oscillating counterpart, the reaction of **8** (pink) to **10** (orange) reaches nearly full conversion whereas that of **15** (teal) to **16** (brick red) reaches a conversion of around 18%. All reactions were carried out in CSTR at 60 °C in DMSO and were monitored using in situ infrared spectroscopy (Supplementary Fig. 34) and [1]H-NMR spectroscopy of samples taken every 30 min.

of oscillations, **8** reaches practically full conversion and **15** reaches a steady state at 18% conversion. Thus, oscillations enhance the mean selectivity by an order of magnitude (Supplementary Table 13).

Simulations predict that the selectivity enhancement depends on [**1**] because the selectivity achieved without oscillations is much smaller for high catalyst concentrations but shows an increase for lower concentrations (Supplementary Fig. 30). We carried out experiments with a lower [**1**] amplitude of 15 ± 3 mM and found that selectivity enhancement dropped fivefold (Supplementary Fig. 36 and Supplementary Table 13). Separate control experiments containing only the Knoevenagel reagents and piperidine catalyst confirm that selectivity similar to that observed in the oscillating system can be achieved at low catalyst loading (5 and 1 mM; Supplementary Fig. 37). By contrast, oscillations allow high selectivity over a large range of catalyst concentrations rather than only at the lower limit. Although conventional catalytic reactions could be optimized in this way, in complex reaction networks in which optimization of single components is not trivial this robustness is an attractive feature.

The modular design of our small-organic-molecule oscillator is a template for oscillators based around different Fmoc-protected amines, provided these are sufficiently basic to perform autocatalytic deprotection. Using amines other than piperidine, especially chiral amines, will further expand the potential of the oscillatory design developed

in this work. The ability to apply a load to our system without the need for extensive reoptimization of its oscillating properties is promising for potential applications, particularly as components in larger chemical reaction networks. Secondary functions in which the oscillating catalyst can be extended include the described ability to maintain high chemoselectivity over a range of catalyst concentrations.

A use case similar in concept to the above-described chemical selectivity enhancement is exploiting the periodic absence of a catalyst or intermediate that is harmful for downstream processes, allowing the performance of incompatible processes within the same system and/or prevention of degradation of reaction products by the catalyst. For instance, biological oscillators are proposed to boost metabolic efficiency in this manner[39,40]. Examples of product degradation that could be avoided in this manner include epimerization of newly formed chiral centres[41] or the formation of the undesired thermodynamic versus the desired kinetic product. A catalytic oscillator can also be considered a catalyst that switches autonomously between an 'on' and 'off' state, whereas reported examples of switchable catalysts that have been used to make copolymers with precisely controlled sequences require external stimuli for switching between active and inactive states or between the production of two separate products[42,43]. Coupling a catalytic oscillator to the synthesis of polymers is therefore another interesting application with potential for modulation of

polymer composition by tuning the characteristics of the oscillator. Periodic catalysis could also be used to drive molecular machines in a time-controlled manner by having the machine perform a function when the catalyst is present and resetting when it is absent[44–47].

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

## Data availability

Experimental procedures and the scripts used to determine at what conditions oscillations can take place, as well as tables with the results of this exploration, are available within the Supplementary Information. The data supporting the findings of this study are available on *DataverseNL* with identifier https://doi.org/10.34894/VPG7MB. Source data are provided with this paper.

**Acknowledgements** We thank the Center for Information Technology of the University of Groningen for their support and for providing access to the Peregrine high-performance computing cluster; the Ministry of Education, Culture and Science for financial support (S.R.H., M.t.H., W.T.S.H. and O.R.M., Gravity programme no. 024.001.035); The Netherlands Organization for Scientific Research (nos. NWO-VICI 724.017.003 to S.R.H. and NWO-Veni 202.155 to A.S.Y.W.); the Simons Collaboration on the Origins of Life (W.T.S.H. and O.R.M., SCOL award no. 477123); and the European Research Council (M.t.H. and S.A.R., ERC grant no. 773264 to S.R.H.).

**Author contributions** S.R.H. directed research and acquired project funding. S.R.H. and M.t.H. conceived the project. M.t.H. designed and developed the catalytic oscillator. A.S.Y.W. consulted regarding models describing organic oscillators. M.t.H., O.R.M., W.T.S.H. and S.R.H. conceptualized flow oscillations. M.t.H. and O.R.M. developed oscillations in flow. S.R.H. and M.t.H. conceptualized oscillating catalysis. M.t.H. developed the Knoevenagel oscillator. S.A.R. developed the Claisen–Schmidt/Michael cascade pulse. S.R.H., M.t.H. and S.A.R. conceptualized oscillation-enhanced selectivity. M.t.H. administrated the project, visualized experiments and wrote the original draft. All authors contributed to reviewing and editing the manuscript.

**Competing interests** The authors declare no competing interests.

**Additional information**
**Correspondence and requests for materials** should be addressed to Wilhelm T. S. Huck or Syuzanna R. Harutyunyan.
