## [Peer Review File · Nature]

Manuscript Title: A catalytically active small organic molecule oscillator

Reviewer Comments & Author Rebuttals

Reviewer Reports on the Initial Version:

Referees' comments:

Referee #1 (Remarks to the Author):

The manuscript describes an autocatalytic chemical reaction network that can generate oscillations of product concentrations in flow. One of the products is a low molecular weight organic amine that is demonstrated to function as an organocatalyst. As the concentration of amine oscillates in time in the flow reactor, so does the amount of product from the organocatalyzed reaction. The major advance of the described system over the state of the art is that a product from the oscillating reaction network is used to carry out a certain secondary function (in this case catalysis), with the oscillation carrying through in that function. Because the amine acts as a catalyst in the secondary function, it is not consumed and therefore the secondary function does not perturb the primary oscillatory network. This is a very clever design choice in the system. Orthogonality of all reactions is essential to the functioning of the full system. This means that all species added or generated should only interact with those parts of the oscillator that they were intended to do. At first sight, this orthogonality seems optimal, but not all possible controls are described. As an example, the reaction of piperidine with the two phenylacetates will generate phenols of varying acidity. To what degree do these phenols interfere with the organocatalyzed reaction, i.e. might they protonate the amine or shift the protonation equilibrium of the malonate? The data on the combined system (Figure 4) seems to show that there is little interference, at least at the chosen conditions. Still, the combined system is demonstrated at basically one set of parameters. Also, Figure 4b is suggested to show a sustained oscillation of the Knoevenagel product. But closer inspection of the orange data gives the impression that there may be multiple regimes: after a first oscillation there are three oscillations with a slight dampening followed by a switch to a truly sustained oscillation. Is this a real effect (and if so, what's causing it?) or is it an artifact or something that is within the noise of the measurement? The general concept of the paper will be of great value to the field of out of equilibrium science and possibly in synthetic life. However, the concept is demonstrated on one, not particularly useful, reaction at one set of reaction parameters that remain constant. As such, it seems a missed opportunity to show the full potential and implications of this magnificent concept. How can this be applied? How sensitive is it to these parameters? What would happen if some parameters are changed during the reaction? Can the secondary function, the catalyzed reaction, respond autonomously to perturbations in the parameter set, which may be used as sort of a process control function as is seen in biological reaction networks? In the same light, in the abstract, the authors state "Autonomous time-controlled chemical synthesis may in future benefit various applications in systems chemistry, automated synthesis, and periodic drug delivery." but what this means is not explained nor are any of these claims demonstrated in the paper.

Minor points:

The oscillator network consists of an autocatalytic Fmoc deprotection to give piperidine, coupled to an acylation reaction. The autocatalytic Fmoc deprotection was pioneered by Ichimura (ref 27) and was also proposed for use in autocatalysis by Semenov (ChemSystemsChem 2020, 2, e2000026, not cited).

Figure S5 (right) shows a linear model fit through 4 datapoints. The fit is of intermediate quality, and the data appears to make a better fit to a nonlinear model. As such, I wonder how strong this evidence is to support the reaction being first order in [Fmoc-pip 2].

For the sake of completeness, it would be good to make the reaction equations such as in Figures 2 or 3 consistent (i.e. show all product including the phenols), if only in the SI.

Referee #2 (Remarks to the Author):

Harutyunyan, Huck, Wong and collaborators present a very nicely designed and characterised oscillator which showcases the benefits of organocatalytic components.

The manuscript is interesting because of the way it solidifies and clarifies the design principles set out in previous papers (ref 18, 21, 22 etc) on how oscillatory behaviour can be observed in reactors. The components making up the systems are similar: a fast inhibitor causing a lag period initially, a self-replicator causing exponential growth after initiation by a trigger molecule, and a slow inhibitor causing eventual decay. The chemical nature does differ from the previous studies, with the autocatalysis in this work depending on basicity.

A rather nice aspect of this work is how the course is charted from optimising a batch reaction, extracting the kinetics of the constituent reactions, using those to perform numerical simulations and then extracting optimal parameters for the stirred tank reaction (to create actual oscillations). It makes the different elements of designing and optimising the system very clear and easy to follow. Most of the manuscript is very well written - in particular the logical reasoning shown and the discussions around parameter choices. The observation that increasing the concentration of the slow inhibitor increases both the amplitude and the period (line 180) could have been slightly better explained and some of the discussion about the Knoevenagel condensation (lines 189-195) don't read very well. Figures 2, 3 and 4 repeat a lot of the same information in the reaction scheme portion, which may be unnecessary. Some key amendments also need to be made to the descriptions of the (batch) system as an oscillator and some consideration given to the sustained/dampened nature of the oscillator.

The oscillating system doesn't seem to actually generate much complexity itself and relies upon continuous flow to produce the most interesting aspects of what it does – the out of equilibrium element and periodic nature. Showing an orthogonal catalytic reaction must not have been practically trivial and it seems to work well, but I think some expressions of what might be achieved with this temporal control are missing. I would also argue that the capabilities of the system are quite singular – it can vary catalyst concentrations – and all of the interesting (future) stuff seems like it would emerge from known catalyst reactivity. I don't feel that given the other systems that have been shown to oscillate in such reactors that this work has a massive wow factor.

Suggestions:

The caption of Fig 2 could benefit from slightly more detail about the dashed lines/model – even just a signpost to more detail later on.

Consider labelling on Fig 2b the points at which nitro-phenyl acetate 3 runs out, for absolute clarity of the triggering events. This will help it match the text below describing the phases. “See arrow 1, Fig 2b... etc”

The comment about the polarity of the solution due to phenyl acetate (line 187) is a little out of the blue and makes the reader feel as if something is missing from the earlier intro and results. It seems like there may be an interesting observation or a potential limitation of the system here. Please expand upon this point or make reference to some discussion in the SI.

When describing the CSTR on lines 137-138, “fresh starting material is continuously added”, it could be beneficial to some readers to explicitly say whether this means all of the reaction components are being added.

Line 193: “Dimethyl malonate (9) is only mildly acidic, and will therefore not inhibit the positive feedback.” is a little isolated and may not fully make sense to some – I presume that it means that 9 will not behave as an acid or base in such a way as to interfere with the oscillator.

Terminology:

Because batch and flow setups are both demonstrated I feel it is important to keep an eye on the usage of ‘oscillation’ as it’s possible to cause confusion about whether or not out-of-equilibrium conditions, which permit oscillations, are possible in batch using this system – genuine oscillations are shown only under continuous flow conditions in this manuscript. Further, oscillations are an inherently non-equilibrium phenomenon and as the authors point out (lines 135-137), their use of a flow CSTR produces the necessary out-of-equilibrium conditions in comparison to equilibrium batch conditions. This mid section of the manuscript surrounding Figs 2 and 3 should be checked and reworded so as to clarify this difference between batch and flow, non-oscillating systems and oscillating ones.

For example: it is not appropriate to describe a single run through Phases I, II and III as a ‘single oscillation’ (line 94 onwards). If there’s no periodicity/repeated events then it is not, by definition, an oscillation. Usages of ‘oscillation’ in this batch context must therefore be altered to something like growth-decay cycles or reaction cycle (or similar).

The authors contrast the continuous flow oscillations to the batch reactivity by describing them as ‘sustained’ oscillations. Related to the reasons above, this comparison can be misleading because it implies that the difference between sustained and non-sustained oscillations is repeated events or periodicity (lines 135, 136, 143). The use of the word ‘sustained’ has been used in literature describing oscillators to mean that there is not a dampening effect on the amplitude of the oscillations and this is factored in by the authors: “...oscillation with no decrease in amplitude.” (line 144). The more accurate and helpful comparison is to state that using a CSTR vs batch, i) actual oscillations can now be produced and ii) that these are sustained oscillations.

Referee #3 (Remarks to the Author):

The authors have constructed a new organic chemical oscillator involving autocatalytic piperidine production and acetylation, and used the output to periodically catalyze formation of a desired product. Although conceptually simple, it was likely a challenge to find a secondary process that operated without interfering with the oscillatory system. This “organocatalytic oscillator” is a big step forward in the design of bioinspired chemical reaction networks. The manuscript is well written and the key points are succinctly presented.

To increase significance, it might also be useful to speculate on the potential functionality of periodic catalysis in a synthetic organic system such as this. With regards to biological oscillators, there has been discussion about increased efficiency of a metabolic process for example, or minimizing exposure of downstream products or catalysts (enzymes) to harmful intermediates, see eg:

- Chandra, Buzi, Doyle Science, 2011
- Moller, Hauser, Olsen Biophys Chem 1998;72:63-72

Can the Knoevenagel condensation reaction be thought of in such a context?

The authors could probably add a reference to note that the periodic production of malonyl and bromous radicals in the BZ reaction was used to catalyze acrylonitrile polymerization, resulting in periodic polymerization. However, bromine also contaminated the product in that case:

- Washington, West, Misra and Pojman J Am Chem Soc 1999, 121, 32, 7373.

The work of Epstein where a core pH oscillator was used to periodically drive a precipitation process might also be mentioned:

- Kurin-Czorgi, Epstein, Orban Nature 2005 Jan 13;433(7022):139-42

So there is some precedent along the lines the authors are developing here, however it is true that this has not been achieved in a purely organic oscillator with a useful product. The system appears reasonably robust and there was no unwanted cross-talk. It may be difficult to modulate this oscillator however this remains to be seen.

Fig 2 – This isn’t usually referred as a “batch oscillation” since there is neither a complete cycle in all species nor repetition of the cycle. As there is a lag phase (or induction period), sometimes this type of behavior is called a clock, but biologists use clock to mean oscillations so this term can also be confusing. The term pulse is probably better.

Fig 3. Increasing the concentration of slow inhibitor phenyl acetate from point g through to h on the phase diagram appears to result in slightly larger amounts of N-Fmoc accumulating, which makes sense. Any particular reason why the authors don’t include examples of oscillations in the model in the SI, pg 21 – is the transition qualitatively the same?

Did the authors monitor the temperature in time, or determine if temperature affects the oscillations dramatically? It’s mentioned in the SI that the bath was kept at 70 C to ensure a temperature of 60 C. How do the authors know that? Also, temperature compensation is an interesting phenomena in biological systems that might be useful in periodic catalysis.

Minor points

Fig 1 – I’m not sure why there is such a large phase lag between catalyst and product P in the sketch? Fig 3b. The writing is particularly difficult to read in this subfigure.

Author Rebuttals to Initial Comments:

General remarks

First of all, we would like to thank the Reviewers for their positive feedback regarding our oscillating catalysis concept. We also would like to acknowledge that as a result of the extensive work directed to addressing all the comments raised by the Reviewers, the manuscript has been improved significantly.

Importantly, we were able to substantiate the potential of our oscillating catalysis concept by introducing *oscillation induced chemoselectivity in catalytic processes* as a promising application, something that would not be possible with conventional catalysis. Furthermore, in this revised version we show three examples of catalyzed reactions coupled to our oscillator. Two additional examples have been demonstrated successfully for pulse in batch compared to the previous version of the manuscript.

We have also studied the sensitivity and robustness of our oscillating catalysis in detail by mapping out its parameter space. Specifically, we have investigated the effects of: a) *the in situ generated phenols*; b) *temperature variations*; c) *the concentration variations of Fmoc-piperidine*; d) *the concentration variations of the slow inhibitor*; e) *the concentration variations of the fast inhibitor*.

Following the request of the Reviewers, we have also added a more substantiated discussion about the possible impact of oscillating catalysis at the end of the manuscript.

Finally, fully sharing the enthusiasm of Reviewers 1 and 3 regarding its potential impact, we would like to emphasize an important additional message of this work: *by rational design one can achieve periodic catalysis of a specific reaction, by taking a known catalyst and turning it into an oscillator*. Moreover, we believe that this work is also an important advance for the emerging field of artificial *on/off switchable catalysis*, as it offers *autonomous on/off catalysis*. As such, this work is also in striking contrast with all examples reported so far in this field, which require external stimuli to switch between the two states, with typically only modest differences in catalytic activity between the 'on' and 'off' states.

More extensive discussion to address the Reviewers comments can be found below.

Referee #1 (Remarks to the Author):

The manuscript describes an autocatalytic chemical reaction network that can generate oscillations of product concentrations in flow. One of the products is a low molecular weight organic amine that is demonstrated to function as an organocatalyst. As the concentration of amine oscillates in time in the flow reactor, so does the amount of product from the organocatalyzed reaction. The major advance of the described system over the state of the art is that a product from the oscillating reaction network is used to carry out a certain secondary function (in this case catalysis), with the oscillation carrying through in that function. Because the amine acts as a catalyst in the secondary function, it is not consumed and therefore the secondary function does not perturb the primary oscillatory network. This is a very clever design choice in the system. Orthogonality of all reactions is essential to the functioning of the full system. This means that all species added or generated should only interact with those parts of the oscillator that they were intended to do.

At first sight, this orthogonality seems optimal, but not all possible controls are described. As an example, the reaction of piperidine with the two phenylacetates will generate phenols of varying acidity.

1. To what degree do these phenols interfere with the organocatalyzed reaction, i.e. might they protonate the amine or shift the protonation equilibrium of the malonate? The data on the combined system (Figure 4) seems to show that there is little interference, at least at the chosen conditions. Still, the combined system is demonstrated at basically one set of parameters.

To understand the effect of the in situ generated phenols on the coupled catalytic reaction, we carried out isolated Knoevenagel condensations in the presence of phenols (phenol, and *p*-nitrophenol). We found that phenol (**14**) had no effect on the condensation and that *p*-nitrophenol (**12**) had a, very mild, inhibitory effect. (Fig. S33)

In response to the concerns of the reviewer, echoed in the comments of Reviewers 2 and 3 as well, we have studied the catalytic oscillator in different conditions (Fig. 4) by varying the concentrations of Fmoc-piperidine, the slow inhibitor and the fast inhibitor (for the latter see SI Fig. S35), and by changing the temperature (Fig. 4 h-i). These experiments show that performing the catalytic reaction does not significantly change the characteristics of the original oscillator, although in some cases it does slightly reduce the oscillatory regime, resulting in dampened oscillations. We also performed an experiment where after two pulses the temperature was raised from 60 °C to 70 °C for 30 minutes (Fig. 4j) and found that this caused the exponential phase to occur sooner, thereby shortening the period of the oscillation. However, once the perturbation had ended the oscillation quickly returned to its normal period.

2. Also, Figure 4b is suggested to show a sustained oscillation of the Knoevenagel product. But closer inspection of the orange data gives the impression that there may be multiple regimes: after a first oscillation there are three oscillations with a slight dampening followed by a switch to a truly sustained oscillation. Is this a real effect (and if so, what's causing it?) or is it an artifact or something that is within the noise of the measurement?

The reviewer is correct in observing that there are two regimes. The oscillator takes a few cycles to settle into its sustained regime. If we plot the absorbance at 784 cm⁻¹ (characteristic of DBF) and at 1566 cm⁻¹ (characteristic of the coumarin) we can see that the system performs a few pulses before settling into a regime of sustained oscillations (i.e. the limit cycle) (Fig. S37).

To test if the system will always perform 3 non-limit cycle pulses before settling into sustained oscillations, a repeat experiment was performed (some points in the first oscillation had to be removed due to a bubble on the IR probe) which showed a similar, albeit not identical, route into the limit cycle (Fig. S37). The initial slight dampening is therefore a real effect but the extent of it varies over repeat experiments. These minor changes are likely caused by noise in the experimental conditions.

3. The general concept of the paper will be of great value to the field of out of equilibrium science and possibly in synthetic life. However, the concept is demonstrated on one, not particularly useful, reaction at one set of reaction parameters that remain constant. As such, it seems a missed opportunity to show the full potential and implications of this magnificent concept. How can this be applied? How sensitive is it to these parameters? What would happen if some parameters are changed during the reaction? Can the secondary function, the catalyzed reaction, respond autonomously to perturbations in the parameter set, which may be used as sort of a process control function as is seen in biological reaction networks?

We fully share the enthusiasm of the reviewer about the potential impact of the work and would like to emphasize an important additional message of this work: by rational design one can achieve periodic catalysis of a specific reaction, by taking a known catalyst and turning it into an oscillator. In our case we are restricted to the reactions that are known to be catalyzed by

piperidine. However, we have now shown three examples (two more than in the earlier version of the manuscript have been demonstrated successfully for pulse in batch): two types of Knoevenagel reactions using either salicyl aldehyde (original version) or benzaldehyde (new, in the revised version) and a Claisen–Schmidt/Michael cascade condensation of formaldehyde with Hagemann's ester (new, in the revised version). While simple condensation of an aldehyde to malonate is indeed not particularly useful, our main model reaction, the Knoevenagel reaction we have demonstrated in flow oscillations, leads to coumarin derivatives which are known for their biological activities. For other types of reactions, an oscillator using a different core structure known for its catalytic activity would have to be designed, following the example presented in this work.

We believe that this work is also an important advance for the field of artificial on/off switchable catalysis, which has been progressively emerging over recent years (see e.g. the review on Artificial switchable catalysts by D. Leigh, [10.1039/C5CS00096C](https://doi.org/10.1039/C5CS00096C)). In contrast to our system, which offers autonomous on/off catalysis, all the examples reported in this field so far require external stimuli to switch between the two states and in the majority of cases there are merely modest differences in the catalytic activities between the ‘on’ and ‘off’ states of the catalysts.

To address the comments about the sensitivity of the system to various parameters and the effect of system perturbations we have performed several experiments, mentioned in the response to comment 1.

Regarding potential and implications: we believe that oscillating catalysis will have interesting applications not possible with conventional catalysis, nor with any other oscillating system reported so far. As a proof of principle, we demonstrate in this revised version that chemoselectivity can be achieved with oscillating catalysis, which is impossible with conventional catalysis (Fig. 5). As this is just one example, and we anticipate that more opportunities will be discovered, an additional discussion about potential applications has been added (lines 315-340).

4. In the same light, in the abstract, the authors state " Autonomous time-controlled chemical synthesis may in future benefit various applications in systems chemistry, automated synthesis, and periodic drug delivery." but what this means is not explained nor are any of these claims demonstrated in the paper.

We have followed the Reviewers’ (1 and 3) suggestion and added a discussion about what might be achieved at the end of the manuscript (lines 315-340).

5. Minor points:

The oscillator network consists of an autocatalytic Fmoc deprotection to give piperidine, coupled to an acylation reaction. The autocatalytic Fmoc deprotection was pioneered by Ichimura (ref 27) and was also proposed for use in autocatalysis by Semenov (ChemSystemsChem 2020, 2, e2000026, not cited).

The citation has been added.

Figure S5 (right) shows a linear model fit through 4 datapoints. The fit is of intermediate quality, and the data appears to make a better fit to a nonlinear model. As such, I wonder how strong this evidence is to support the reaction being first order in [Fmoc-pip 2].

We added initial rates measured at 50 mM and 60 mM Fmoc-piperidine to strengthen the linear trend. We find that the trend is linear over a wider range of concentrations, showing that the reaction is 1st order in Fmoc-piperidine (2) (Fig. S5).

For the sake of completeness, it would be good to make the reaction equations such as in Figures 2 or 3 consistent (i.e. show all product including the phenols), if only in the SI.

The phenols have been added to the figures in the main text, and to those in the SI where appropriate.

Referee #2 (Remarks to the Author):

Harutyunyan, Huck, Wong and collaborators present a very nicely designed and characterised oscillator which showcases the benefits of organocatalytic components.

The manuscript is interesting because of the way it solidifies and clarifies the design principles set out in previous papers (ref 18, 21, 22 etc) on how oscillatory behaviour can be observed in reactors.

The components making up the systems are similar: a fast inhibitor causing a lag period initially, a self-replicator causing exponential growth after initiation by a trigger molecule, and a slow inhibitor causing eventual decay. The chemical nature does differ from the previous studies, with the autocatalysis in this work depending on basicity.

A rather nice aspect of this work is how the course is charted from optimising a batch reaction, extracting the kinetics of the constituent reactions, using those to perform numerical simulations and then extracting optimal parameters for the stirred tank reaction (to create actual oscillations). It makes the different elements of designing and optimising the system very clear and easy to follow.

1. Most of the manuscript it very well written - in particular the logical reasoning shown and the discussions around parameter choices. The observation that increasing the concentration of the slow inhibitor increases both the amplitude and the period (line 180) could have been slightly better explained and some of the discussion about the Knoevenagel condensation (lines 189-195) don't read very well.

The text was rewritten to better explain these observations.

2. Figures 2, 3 and 4 repeat a lot of the same information in the reaction scheme portion, which may be unnecessary. Some key amendments also need to be made to the descriptions of the (batch) system as an oscillator and some consideration given to the sustained/dampened nature of the oscillator.

The schemes can be removed from the figures if required, but we would prefer to keep all the information required to easily understand all panels in one cohesive figure. We will defer to the decision of the editor for the final manuscript.

3. The oscillating system doesn't seem to actually generate much complexity itself and relies upon continuous flow to produce the most interesting aspects of what it does – the out of equilibrium element and periodic nature. Showing an orthogonal catalytic reaction must not have been practically trivial and it seems to work well, but I think some expressions of what might be achieved with this temporal control are missing. I would also argue that the capabilities of the system are quite singular – it can vary catalyst concentrations – and all of the interesting (future)

stuff seems like it would emerge from known catalyst reactivity. I don't feel that given the other systems that have been shown to oscillate in such reactors that this work has a massive wow factor.

As already mentioned in the comment to Reviewer 1, we believe that oscillating catalysis will have interesting applications not possible by conventional catalysis and therefore also not with the oscillating systems reported so far. As a proof of principle, we demonstrate in this revised version that chemoselectivity between two very structurally similar reagents can be achieved with oscillating catalysis (Fig. 5), which is impossible with conventional catalysis. As this is just one example, and we anticipate that more opportunities will be discovered, an additional discussion about potential applications has been added (lines 315-340).

4. *Suggestions:*

The caption of Fig 2 could benefit from slightly more detail about the dashed lines/model – even just a signpost to more detail later on.

A more detailed description of the dashed lines was added to the caption.

Consider labelling on Fig 2b the points at which nitro-phenyl acetate 3 runs out, for absolute clarity of the triggering events. This will help it match the text below describing the phases. “See arrow 1, Fig 2b... etc”

Arrows were added to figure 2b and referenced in the discussion of the figure in the main text.

The comment about the polarity of the solution due to phenyl acetate (line 187) is a little out of the blue and makes the reader feel as if something is missing from the earlier intro and results.

This section was rewritten. We suspect that increased concentration of phenylacetate will change the polarity of the reaction media and consequently the rates of the involved reactions will be affected. We are currently busy measuring the rates of the reactions at different concentrations of phenyl acetate. These results will be reported in follow up work.

When describing the CSTR on lines 137-138, “fresh starting material is continuously added”, it could be beneficial to some readers to explicitly say whether this means all of the reaction components are being added.

To remove the ambiguity the text now states explicitly that all reaction components are being added.

Line 193: “Dimethyl malonate (9) is only mildly acidic, and will therefore not inhibit the positive feedback.” is a little isolated and may not fully make sense to some – I presume that it means that 9 will not behave as an acid or base in such a way as to interfere with the oscillator.

This sentence was removed as we agree it contains too much technical detail.

5. *Terminology:*

Because batch and flow setups are both demonstrated I feel it is important to keep an eye on the usage of ‘oscillation’ as it's possible to cause confusion about whether or not out-of-equilibrium conditions, which permit oscillations, are possible in batch using this system – genuine oscillations are shown only under continuous flow conditions in this manuscript. Further, oscillations are an inherently non-equilibrium phenomenon and as the authors point out (lines

135-137), their use of a flow CSTR produces the necessary out-of-equilibrium conditions in comparison to equilibrium batch conditions. This mid-section of the manuscript surrounding Figs 2 and 3 should be checked and reworded so as to clarify this difference between batch and flow, non-oscillating systems and oscillating ones. For example: it is not appropriate to describe a single run through Phases I, II and III as a 'single oscillation' (line 94 onwards). If there's no periodicity/repeated events then it is not, by definition, an oscillation. Usages of 'oscillation' in this batch context must therefore be altered to something like growth-decay cycles or reaction cycle (or similar).

The words "batch oscillation" were replaced with "pulse" or "pulse in batch" in the manuscript and SI as recommended by reviewer #3.

The authors contrast the continuous flow oscillations to the batch reactivity by describing them as 'sustained' oscillations. Related to the reasons above, this comparison can be misleading because it implies that the difference between sustained and non-sustained oscillations is repeated events or periodicity (lines 135, 136, 143). The use of the word 'sustained' has been used in literature describing oscillators to mean that there is not a dampening effect on the amplitude of the oscillations and this is factored in by the authors: "...oscillation with no decrease in amplitude." (line 144). The more accurate and helpful comparison is to state that using a CSTR vs batch, i) actual oscillations can now be produced and ii) that these are sustained oscillations.

The wording of this section was changed to improve clarity. Additionally, the area where dampened oscillations are predicted has been added to figures 3 and 4.

Referee #3 (Remarks to the Author):

The authors have constructed a new organic chemical oscillator involving autocatalytic piperidine production and acetylation, and used the output to periodically catalyze formation of a desired product. Although conceptually simple, it was likely a challenge to find a secondary process that operated without interfering with the oscillatory system. This "organocatalytic oscillator" is a big step forward in the design of bioinspired chemical reaction networks. The manuscript is well written and the key points are succinctly presented.

1. To increase significance, it might also be useful to speculate on the potential functionality of periodic catalysis in a synthetic organic system such as this. With regards to biological oscillators, there has been discussion about increased efficiency of a metabolic process for example, or minimizing exposure of downstream products or catalysts (enzymes) to harmful intermediates, see eg:

- Chandra, Buzi, Doyle Science, 2011
- Moller, Hauser, Olsen Biophys Chem 1998;72:63-72

Can the Knoevenagel condensation reaction be thought of in such a context?

This is a great suggestion. We have added it to the discussion of potential applications of catalytic oscillators (lines 315-340). In addition, as mentioned in the response to Reviewers 1 and 2, as a proof of principle, we demonstrated that with oscillating catalysis chemoselectivity

can be achieved, which is not possible with conventional catalysis (Fig 5). This is just one example, and we anticipate to have more experimental examples in follow up work.

2. The authors could probably add a reference to note that the periodic production of malonyl and bromous radicals in the BZ reaction was used to catalyze acrylonitrile polymerization, resulting in periodic polymerization. However, bromine also contaminated the product in that case:

- Washington, West, Misra and Pojman J Am Chem Soc 1999, 121, 32, 7373.

This reference has been added to the introduction in the following paragraph: “Of the known oscillators only the Belousov-Zhabotinsky reaction has ever been used to control another reaction, but in that case the product of the reaction was severely modified by the components of the chemical oscillator.”

3. The work of Epstein where a core pH oscillator was used to periodically drive a precipitation process might also be mentioned:

- Kurin-Czorgi, Epstein, Orban Nature 2005 Jan 13;433(7022):139-42

This work was mentioned in the summation of oscillator systems as: “an oscillator aimed at probing biological systems”.

4. So there is some precedent along the lines the authors are developing here, however it is true that this has not been achieved in a purely organic oscillator with a useful product. The system appears reasonably robust and there was no unwanted cross-talk. It may be difficult to modulate this oscillator however this remains to be seen.
5. Fig 2 – This isn’t usually referred as a “batch oscillation” since there is neither a complete cycle in all species nor repetition of the cycle. As there is a lag phase (or induction period), sometimes this type of behavior is called a clock, but biologists use clock to mean oscillations so this term can also be confusing. The term pulse is probably better.

The words “batch oscillation” were replaced with “pulse” and “pulse in batch”.

6. Fig 3. Increasing the concentration of slow inhibitor phenyl acetate from point g through to h on the phase diagram appears to result in slightly larger amounts of N-Fmoc accumulating, which makes sense. Any particular reason why the authors don’t include examples of oscillations in the model in the SI, pg 21 – is the transition qualitatively the same?

The predicted oscillations at the tested conditions were included in the SI as figure S31. The transition is not qualitatively the same as the model had predicted no significant increase in the amount of N-Fmoc accumulating upon increasing the concentration of phenyl acetate from 1.4 M to 2.2 M.

7. Did the authors monitor the temperature in time, or determine if temperature affects the oscillations dramatically? It’s mentioned in the SI that the bath was kept at 70 C to ensure a temperature of 60 C. How do the authors know that? Also, temperature compensation is an interesting phenomena in biological systems that might be useful in periodic catalysis.

The internal temperature was monitored using the IR thermometer incorporated in the in situ IR probe. This information has been added to the description of the experiments in the SI. Temperature compensation would indeed be an interesting phenomenon to explore further. Unfortunately, we have seen no sign of it in our oscillator.

8. Minor points

Fig 1 – I'm not sure why there is such a large phase lag between catalyst and product P in the sketch?

The phase lag in the figure has been slightly reduced and the labels have been changed to be clearer. We want to have some phase lag in the figure because the product of the catalytic reaction will always lag slightly behind the catalyst itself, because the catalyzed reaction occurs on a finite, non-zero timescale.

Fig 3b. The writing is particularly difficult to read in this subfigure.

The size of the graphs (and thus the text) has been increased by 10 percent.

Reviewer Reports on the First Revision:

Referees' comments:

Referee #1 (Remarks to the Author):

The authors have done a great job addressing the remarks of the referees and this has resulted in a greatly improved paper, which is really on a higher level now. I'm also intrigued by the prospect of achieving chemoselectivity through oscillatory catalysis, as was introduced in this revision. If true, this is a fascinating advance. A problem is that I simply don't understand it. How can it be that it matters if the concentration of a catalyst oscillates in time, that it leaves a substrate alone when oscillating but it reacts when it doesn't oscillate? In all oscillatory cases (batch oscillations and flow), shortly after reaching the DBF peak, there should still be a considerable amount of catalyst present. At that moment, the salicyl aldehyde concentration is very low (it has reacted away already). Why doesn't the benzaldehyde react then? Shortly after the start of every process (either oscillating or not), there will be a large amount of active catalyst. I would think this catalyst will react with the various substrates purely based on the ratio of rate constants, but that doesn't seem to be the case. It's quite possible that I'm missing the point somewhere, but perhaps it would be useful if this phenomenon is either explained better, or is supported by some sort of simulation demonstrating it.

minor point:

The NMR spectra in the SI have a grid superimposed. For single spectra this is unnecessary but not a problem. For stacked spectra, and especially offset stacked spectra, this grid makes the data very difficult to digest, and should be removed.

Referee #2 (Remarks to the Author):

I think the manuscript is much improved and feel that the additional experiments have allowed the authors to flush out what is conceptually interesting in meaningful way.

Just a couple of things - if referee 1 is correct about the systems design being previously suggested by Semenov, then proper credit needs to be given, not the way it is being done now.

The data for the oscillator selectivity never actually shows the concentration of products formed. In the main text and figure 5 just shows the disappearance of SM, with a reference to the SI for batch experiments. The SI S36 shows a batch run with the disappearance of the SMs, but formation of products is not plotted. On pg 32 there are some plots of NMRs in time, but the times on the axis are discontinuous somehow so that it isn't actually obvious what is going on, and I can't see a signal for 16.

The selective formation of products during oscillations needs to be shown somewhere, and this should be in the Figure 5 with supporting data somewhere in the SI.

9 and 15 may appear to only be trivially different because of the substitution pattern, but the properties of the two phenols would not be expected to be the same at all; Hammett parameters for example don't use ortho substituents because the proximity to the reactive centre complicates things. 9 could disappear before 15 in lots of ways besides actually giving products, including

reversible addition of water or the catalyst.

Referee #3 (Remarks to the Author):

The authors have made considerable improvements to the manuscript with the addition of plots demonstrating the robustness of the oscillator with additional catalysis and potential for innovative features such as selective catalysis in the presence of multiple substrates. This one-pot approach may be useful for separations.

Minor points:

Actually the mean period looks slightly smaller in all cases with catalysis than without; a statistical comparison might be useful to be able to say that “a load can be applied to our oscillator in a catalytic fashion without altering the core characteristics of the oscillator itself”, and later the “catalytic reaction does not change the characteristics of the oscillator” - nevertheless the alteration is a minor one.

With the normalised absorption in Fig 5, we cannot compare the two sets of conditions easily; it would be useful to have an idea of the steady state amount of piperidine produced in the non-oscillating case, compared to the maximum or average amount in the oscillating case. This could be in the supplementary information.

We appreciate all the suggestions made by the Referees to further improve the manuscript and hope that with additional experiments and revisions we were able to address all their concerns.

Specific changes in the revised version in both main text and SI (highlighted in yellow)

- In two places the main text has been revised to present the concept of selectivity through oscillating catalysis more clearly.
- Figure 5a in the main text has been revised as well by adding to the graphs data for coumarin formation.
- Figure S38 with simulations for selectivity through oscillating catalysis has been added to SI.
- Several control experiments (procedures and figure S40) showing products formation from the corresponding aldehydes in different conditions, and the synthesis of a reference sample of **16** have been added to the SI as well.
- Because of added figures and schemes numbering in SI has been modified.

Detailed Response to Referees' comments:

Referee #1 (Remarks to the Author):

The authors have done a great job addressing the remarks of the referees and this has resulted in a greatly improved paper, which is really on a higher level now. I'm also intrigued by the prospect of achieving chemoselectivity through oscillatory catalysis, as was introduced in this revision. If true, this is a fascinating advance. A problem is that I simply don't understand it. How can it be that it matters if the concentration of a catalyst oscillates in time, that it leaves a substrate alone when oscillating but it reacts when it doesn't oscillate? In all oscillatory cases (batch oscillations and flow), shortly after reaching the DBF peak, there should still be a considerable amount of catalyst present. At that moment, the salicyl aldehyde concentration is very low (it has reacted away already). Why doesn't the benzaldehyde react then? Shortly after the start of every process (either oscillating or not), there will be a large amount of active catalyst. I would think this catalyst will react with the various substrates purely based on the ratio of rate constants, but that doesn't seem to be the case. It's quite possible that I'm missing the point somewhere, but perhaps it would be useful if this phenomenon is either explained better, or is supported by some sort of simulation demonstrating it.

We thank the Referee 1 for pointing this out and apologize that we did not explain the concept clearly in our preceding draft. The Referee is correct in stating that when the salicyl aldehyde concentration is low, *p*-hydroxybenzaldehyde reacts as well. However, because the reaction rate of *p*-hydroxybenzaldehyde is much lower, it requires a high catalyst concentration to be converted, as the single-pulse experiment depicted in Figure S39 shows, where only a few percent conversion is reached. During oscillating catalysis, the catalyst concentration is high enough for *p*-hydroxybenzaldehyde to react for only short periods and therefore selectivity can be achieved. On the contrary, in a conventional CSTR setup the catalyst is present continuously, therefore the less reactive substrate can be freely converted after the more reactive substrate has reached full conversion. The confusion comes from Figure 5a where it seems like *p*-hydroxybenzaldehyde is not being converted at all. This is due to the experimental limitations we have with the measurements for the selectivity experiment in flow: continuous NMR measurements are not possible and the *in-situ* IR monitoring of the reaction progress was only possible for **DBF** because the IR signals for the aldehydes and their products overlap. Therefore, these concentrations had to be monitored by taking samples followed by NMR measurements, which is why we have coarser time sampling, not sufficient to detect very small changes in the concentration of *p*-hydroxybenzaldehyde.

Following the Referee's advice, we run simulations for selectivity through oscillatory catalysis (Figure S38). It was found that in the presence of oscillations the fast reaction oscillates between 40% and 80% conversion while the slow reacting substrate oscillates between 0.5% and 2.5% conversion. When the

simulation is run without phenyl acetate (4), and oscillations are not occurring, the fast substrate reaches full conversion and the slow substrate reaches 50% conversion. These simulations demonstrate that using chemical oscillations selectivity can be achieved.

Figure S38 Simulations demonstrating the concept of oscillation induced selectivity. *a*, Reaction scheme showing the conditions used for the simulations. *b*, Simulation of oscillations inducing selectivity. Traces are shown for the concentrations of piperidine (green), substrate 1 (purple), and substrate 2 (teal). Piperidine oscillates between 0 and 18 mM. *c*, Simulation showing that without oscillations both substrates are converted. Traces are shown for the concentrations of piperidine (green), substrate 1 (purple), and substrate 2 (teal). Piperidine reaches a steady state concentration of 70 mM.

In the manuscript, the following textual changes have been made to present the concept more clearly:

The explanation of the concept has been expanded (Lines 273-278).

In an oscillator the catalyst is only available for short periods. During such a pulse the more reactive substrate can reach a high conversion, but the less reactive substrate can reach only a very low conversion before the catalyst disappears. On the contrary, in a conventional CSTR setup in which the catalyst is present continuously, the less reactive substrate can be freely converted after the more reactive substrate has reached full conversion. For a simulation demonstrating this concept see Fig. S38.

Additionally, the experimental results are presented in more detail (Lines 284-291)

We observed that in a single pulse 8 is converted fully, while 15 only reaches 10% conversion and then remains constant because the drop in catalyst concentration halts the reaction (Fig. S39). In the absence of a pulse 8 is converted fully, after which 15 continues to react, reaching 50% conversion after 60 minutes. Performing the competition experiment with an oscillator in flow, it was rewarding to find that while 8 is converted, 15 seems to be left untouched (Fig. 5a). Although some conversion must take place, it is subtle – and not possible to measure reliably – consistent with the simulation in Fig. S38.

minor point:

The NMR spectra in the SI have a grid superimposed. For single spectra this is unnecessary but not a problem. For stacked spectra, and especially offset stacked spectra, this grid makes the data very difficult to digest, and should be removed.

Gridlines have been removed from the stacked spectra.

Referee #2 (Remarks to the Author):

I think the manuscript is much improved and feel that the additional experiments have allowed the authors to flush out what is conceptually interesting in meaningful way.

Just a couple of things - if referee 1 is correct about the systems design being previously suggested by Semenov, then proper credit needs to be given, not the way it is being done now.

The design of an oscillator using Fmoc deprotection has not been previously suggested. The fact that deprotection of Fmoc protected piperidine is autocatalytic was reported for the first time by Ichimura, which we have cited. Semenov's work is a review about all the reported examples of autocatalytic reactions and their mechanisms. Fmoc deprotection is also mentioned as one example of an autocatalytic reaction, with a reference to Ichimura's work. There is, however, no discussion or hypothesis regarding how this specific reaction is useful for oscillating systems or any other system. Therefore, we feel that the work of both Ichimura and Semenov is appropriately credited. But we are happy to consider if the Referee has a specific suggestion.

The data for the oscillator selectivity never actually shows the concentration of products formed. In the main text and figure 5 just shows the disappearance of SM, with a reference to the SI for batch experiments. The SI S36 shows a batch run with the disappearance of the SMs, but formation of products is not plotted. On pg 32 there are some plots of NMRs in time, but the times on the axis are discontinuous somehow so that it isn't actually obvious what is going on, and I can't see a signal for 16. The selective formation of products during oscillations needs to be shown somewhere, and this should be in the Figure 5 with supporting data somewhere in the SI. 9 and 15 may appear to only be trivially different because of the substitution pattern, but the properties of the two phenols would not be expected to be the same at all; Hammett parameters for example don't use ortho substituents because the proximity to the reactive centre complicates things. 9 could disappear before 15 in lots of ways besides actually giving products, including reversible addition of water or the catalyst.

We thank the Referee for their suggestions. Formation of **10** has been added to Figure S39 (previously S36) and to Figure 5 (below), showing that **9** is indeed reacting to form **10** and is not consumed by some other process.

We are unable to include a trace for **16** because the signal overlaps too much with the other molecules present to be measured individually.

To show that both Knoevenagel condensations proceed cleanly to the products we have also included in SI control experiments that demonstrate this (Fig. S40).

Fig. 5. Oscillation induced selectivity. **a**, Oscillation experiment showing selectivity for salicylaldehyde (8) over *p*-hydroxybenzaldehyde (15). Flow experiment carried out in CSTR at 60 °C using 100 mM Fmoc-piperidine (2), 30 mM *p*-nitrophenyl acetate (3), 1.8 M phenyl acetate (4), 5 mM *N*-methylpiperidine (5), 50 mM salicylaldehyde (8), 50 mM *p*-hydroxybenzaldehyde (15), and 200 mM dimethyl malonate (9) in DMSO with a space velocity of 10^{-4} s $^{-1}$. The oscillation was monitored using in situ IR spectroscopy and 1 H-NMR of samples taken every 30 minutes. Sustained oscillations are obtained for DBF (6, red) with a period of 2.22 ± 0.21 hours. The concentrations of salicylaldehyde (8, pink) and 3-(methoxycarbonyl)coumarin (10, orange) were found to oscillate where the concentration of *p*-hydroxybenzaldehyde (15, teal) remains constant. **c** Control experiment performed with the same conditions as **a** but without phenyl acetate (4) showing that without oscillations there is no selectivity.

Referee #3 (Remarks to the Author):

The authors have made considerable improvements to the manuscript with the addition of plots demonstrating the robustness of the oscillator with additional catalysis and potential for innovative features such as selective catalysis in the presence of multiple substrates. This one-pot approach may be useful for separations.

Minor points:

Actually the mean period looks slightly smaller in all cases with catalysis than without; a statistical comparison might be useful to be able to say that “a load can be applied to our oscillator in a catalytic fashion without altering the core characteristics of the oscillator itself”, and later the “catalytic reaction does not change the characteristics of the oscillator” - nevertheless the alteration is a minor one.

To do a proper statistical analysis of the minor alterations observed by the Referee a large number of repeat experiments would be needed. We prefer to rephrase the text indicating that there is only a small influence (as opposed to none).

To better reflect this, the wording has been altered to “a load can be applied to our oscillator in a catalytic fashion without significantly altering the core characteristics of the oscillator itself” and “catalytic reaction does not significantly change the characteristics of the oscillator”.

With the normalised absorption in Fig 5, we cannot compare the two sets of conditions easily; it would be useful to have an idea of the steady state amount of piperidine produced in the non-oscillating case, compared to the maximum or average amount in the oscillating case. This could be in the supplementary information.

Unfortunately, we were not able to accurately determine the concentration of piperidine from the samples taken of the oscillation induced selectivity experiment. As explained before, piperidine itself can't be followed directly due to peak overlap. Unfortunately, also the indirect method of subtracting the concentration of *N*-acetyl piperidine from the concentration of DBF proved insufficient in this case, as the error in the measurement is too large as a result of the 25-fold dilution.

However, at the request of Referee #1 we have included simulations demonstrating the concept of oscillation induced selectivity (Fig. S38). Using these simulations, we can obtain an estimate of the concentration of piperidine in the experiments. During the sustained oscillations the concentration of piperidine oscillates between 0 and 18 mM, while in the absence of oscillations a steady state concentration of 70 mM of piperidine is reached.

Reviewer Reports on the Second Revision:

Referees' comments:

Referee #1 (Remarks to the Author):

The draft addresses the points raised by the referees in a satisfactory manner, leading to an improved manuscript. I think the work described in this paper is a groundbreaking result and I congratulate the authors on the great achievement.

Referee #2 (Remarks to the Author):

I apologise for the length of these comments, but the observation of selection is interesting, but it isn't clear where it arises from and is obviously something that will be of interest to those who read the paper.

Oscillation-induced selectivity descriptions

The following statement in the caption of Fig 5 needs modifying as it implies that all of the selectivity seen emerges from the oscillator.

“Control experiment performed with the same conditions as a but without phenyl acetate (4) showing that without oscillations there is no selectivity”

As shown in Fig S40 (and Fig5b), the substrates 15 and 8 react with 9 at different intrinsic rates. There is inherent selectivity when a mixture is used because different chemical reagents (8, 15) are interacting in different reactions to form different products. It is important not to overstate the selection phenomenon observed and would be better to say that the product selection is amplified (or similar) by the oscillator, because it suppresses the amount of catalyst present.

Similarly, the wording of the sentence on line 292 is slightly ambiguous. It would be better to say that the catalysis is ‘...more selective than without oscillations...’

“Thus, we confirm that oscillating catalysis allows catalytic processes to be more selective in the presence of multiple substrates.”

Catalyst concentration in selection experiments

Referee 3 pointed out that the amount of piperidine catalyst (1) is not comparable between Fig 5a and 5b. The authors state that due to practical limitations, their best estimate for the piperidine concentration in these specific experiments is 0-18 mM (Fig 5a) and 70 mM (Fig 5b).

“During the sustained oscillations the concentration of piperidine oscillates between 0 and 18 mM, while in the absence of oscillations a steady state concentration of 70 mM of piperidine is reached.”

As the reactivity of the p-hydroxybenzaldehyde substrate 15 seems to be sensitive to catalyst concentration (supported by Fig 5b and Fig S40a vs S40b), the large difference in 1 concentration (> factor of 3.8) between the oscillating and non-oscillating experiments (Fig 5) seems a strong reason for the selectivity seen, rather than the presence of oscillations, per se.

As it appears there's a [1] threshold at which the p-hydroxybenzaldehyde doesn't react any further (Fig 5b), wouldn't a non-oscillating system with a fixed but low catalyst concentration (<< 70 mM) achieve the same selectivity as the oscillating system does in Fig 5a? This experiment would be very

informative and if included in the main text could provide more clarity to the oscillation-induced selectivity than the present Fig 5b.

Similarly, if an experiment analogous to that shown in Fig 5a, but with higher oscillating concentrations of catalyst 1 were performed (e.g. between 60-80 mM), it seems likely that the high levels of selection seen in Fig 5a would not be observed and the proportions of 8 and 15 would more likely resemble those in Fig 5b. This would be a key control experiment to determine if the selection is or is not an intrinsic property of the oscillations.

Could the authors clarify what the two horizontal arrows next to the red curve (normalised piperidine absorbance) in both Figures 5a and 5b are indicating? Prior to the comments by Referee 3 about the comparison of normalised absorbance in Fig 5, it was assumed that these arrows indicate comparable concentrations of piperidine in each experiment (Fig 5a and 5b) but this is now potentially misleading. The caption of Fig 5 needs to be explicit as to the fact that the normalised concentrations of piperidine are not comparable between Fig 5a and 5b. Ideally this figure would include the authors estimates for the concentration of 1 in those experiments either in the caption text or the image (0-18 mM and 70 mM).

Quantification and side reactions

There seems to be some reasonable practical obstacles to monitoring the system, particularly in the flow apparatus and this is ascribed to overlapping signals (by NMR and/or IR). However, it isn't well explained why the method of NMR aliquots used for the batch experiments can't be used in the CSTR, can't the outflow material itself be used for analysis without disrupting the experiment? The compound spectra provided in Figs S21-29 are appreciated, but it still isn't clear which signals are preventing the monitoring of the concentration of 16, do the authors mean when using NMR or IR?

The spectra in Figs S21-27 are in CDCl₃ rather than d₆-DMSO. It would be appropriate to provide spectra of these compounds in d₆-DMSO so that comparison of their NMR signals can be made to Fig S19 (and others). This is of particular importance for clarity on whether the signals of 16 and other species overlap in the reaction solvent.

The offset presentation of the stacked NMR data in Figure S19 also does not make it obvious where the overlapping signals that obscure 16 could be. I suggest that offset and non-offset stacked NMR traces are both shown for the reaction timecourse so the obscured signals (or absence of signals from 16) can be better distinguished. According to its NMR spectrum (Fig S28), product 16 has signals at 6.84, 6.82, 3.80 and 3.75 ppm which are close to but sufficiently separated from other compounds' multiplets for some quantification. Can the authors explain whether any quantification of 16 is possible by NMR aliquots of the oscillating CSTR reaction (Figure 5a experiment), specifically using the signals of 16 described above?

In Figure S19 there is the growth of a sizeable multiplet at ~6.8-6.9 ppm but this doesn't obviously correlate to any of the characterised compounds in the mixture. Can the authors comment on what this compound is and whether it affects the system?

Related, the authors explain in response to Referee 3 that the concentration of piperidine could not be determined in the experiments of Fig 5a due to error and dilution of the aliquots. Couldn't the concentration of piperidine and other components (e.g. 16) be readily determined by a more sensitive technique such as LCMS, HPLC or GC?

Have the authors also considered that 16 could be a potential acid-base inhibitor of piperidine (similar to p-nitrophenol inhibition which is acknowledged by the authors)? The hydroxyl of product 16 is expected to be more acidic than that of 15 and its pKa probably resembles p-nitrophenol. This could be an underlying driver for the selection seen and/or why reaction with substrate 15 is so sensitive to catalyst concentration (e.g. large rate deceleration seen in Fig S40b when an equimolar amount of product is formed, relative to catalyst 1). This inhibition is not possible with coumarin product 10 and highlights why forming two different product classes could be problematic when benchmarking selectivity.

Line 233 – Fig S32 should read Fig S34

Fig S38 – in the coloured caption above, both of the substrates are labelled Substrate 1

Fig S38 – the use of both teal and green in the simulation plot makes matching the curves to the coloured compounds in the legend above a little tough, using an alternate colour would make understanding of this figure smoother.

Fig S12 – caption says “CSTR pulse in batch” is this correct? Should this read ‘stirred pulse in batch’? rather than implying flow?

Fig S40a – is the inclusion of p-nitrophenol in the control experiment for the Knoevenagel reaction between 8 and 9 to slow the kinetics for monitoring? Or does it have some other role besides being a weak acid?

Author Rebuttals to Second Revision:

Referee #1 (Remarks to the Author):

The draft addresses the points raised by the referees in a satisfactory manner, leading to an improved manuscript. I think the work described in this paper is a groundbreaking result and I congratulate the authors on the great achievement.

We thank the Referee for all the valuable feedback and suggestions that have improved the manuscript.

Referee #2 (Remarks to the Author):

1. Oscillation-induced selectivity descriptions

The following statement in the caption of Fig 5 needs modifying as it implies that all of the selectivity seen emerges from the oscillator. "Control experiment performed with the same conditions as a but without phenyl acetate (4) showing that without oscillations there is no selectivity" As shown in Fig S40 (and Fig5b), the substrates 15 and 8 react with 9 at different intrinsic rates. There is inherent selectivity when a mixture is used because different chemical reagents (8, 15) are interacting in different reactions to form different products. It is important not to overstate the selection phenomenon observed and would be better to say that the product selection is amplified (or similar) by the oscillator, because it suppresses the amount of catalyst present. Similarly, the wording of the sentence on line 292 is slightly ambiguous. It would be better to say that the catalysis is '...more selective than without oscillations...' "Thus, we confirm that oscillating catalysis allows catalytic processes to be more selective in the presence of multiple substrates."

We appreciate the critical analysis of the Referee. Following his request, we have removed the original statement from the caption of Fig. 5, which now instead states the conversion reached by the substrates in the control experiments.

The main text has been expanded to better explain the selectivity experiments and to introduce additional experiments. Among the rewriting the sentence referred to by the referee has also been rephrased to: "These experiments confirm that oscillating catalysis allows catalytic processes to be more selective than catalysis without oscillations in the presence of multiple substrates." (now line 305-306).

2. Catalyst concentration in selection experiments

Referee 3 pointed out that the amount of piperidine catalyst (1) is not comparable between Fig 5a and 5b. The authors state that due to practical limitations, their best estimate for the piperidine concentration in these specific experiments is 0-18 mM (Fig 5a) and 70 mM (Fig 5b). "During the sustained oscillations the concentration of piperidine oscillates between 0 and 18 mM, while in the absence of oscillations a steady state concentration of 70 mM of piperidine is reached." As the reactivity of the *p*-hydroxybenzaldehyde substrate 15 seems to be sensitive to catalyst concentration (supported by Fig 5b and Fig S40a vs S40b), the large difference in 1 concentration (> factor of 3.8) between the oscillating and non-oscillating experiments (Fig 5) seems a strong reason for the selectivity seen, rather than the presence of oscillations, per se. As it appears there's a [1] threshold at which the *p*-hydroxybenzaldehyde doesn't react any further (Fig 5b), wouldn't a non-oscillating system with a fixed but low catalyst concentration (<< 70 mM) achieve the same selectivity as the oscillating system does in Fig 5a? This experiment would be very informative and if included in the main text could provide more clarity to the oscillation-induced selectivity than the present Fig 5b.

It is a misconception that there is a threshold of [1] where the *p*-hydroxybenzaldehyde is no longer converted. The control experiment of Fig. 5b is carried out in a **flow setup in the absence of phenyl acetate**. Therefore, after a while the system reaches a steady state where the reaction rates match the in- and out-flow of materials (akin to a chemical equilibrium), resulting in constant reactant concentrations. However, control experiments in a batch set up (Fig. S55) show that the *p*-hydroxybenzaldehyde conversion continues as expected, reaching 90% over time.

Following the Referee's request we have made an attempt to estimate experimentally the concentration of piperidine during the oscillations. By running the blank oscillator in DMSO-d₆ and taking twice larger samples with much shorter pauses between samples we were able to obtain an estimate of the piperidine peak concentration (Fig. S54). These experiments were done in duplo. We found that the piperidine amplitude was 15±1 mM, consistent with the predicted 0–18 mM. It should be noted that this is still only an estimate as the previously stated problems regarding precise determination of piperidine concentration remain. Additionally, as we are now taking larger samples the system is perturbed and the experiment has to be stopped after the samples are taken.

We then performed the control experiment in flow requested by the Referee, namely with a piperidine concentration of 15±3 mM to eliminate the difference in concentration from the comparison (Fig. 5b, i). Here we found that **8** is converted fully and **15** reaches 20% conversion, demonstrating that the difference in selectivity between the oscillating and non-oscillating experiments persists and does not arise solely from the difference in concentration of piperidine. Thus, the time that the catalyst is present in the reaction medium strongly affects the obtained result and can therefore be used to obtain selectivity.

Similarly, if an experiment analogous to that shown in Fig 5a, but with higher oscillating concentrations of catalyst **1** were performed (e.g. between 60-80 mM), it seems likely that the high levels of selection seen in Fig 5a would not be observed and the proportions of **8** and **15** would more likely resemble those in Fig 5b. This would be a key control experiment to determine if the selection is or is not an intrinsic property of the oscillations.

Following the request, we increased the concentration of piperidine in the oscillating system, but were not able to obtain concentrations up to 60-80 mM due to limitations on the solubility of the components of the oscillator. The maximum concentration of piperidine we could reach was 22±1 mM (Fig. S54). At these conditions, we still observe selectivity for **8** over **15** (Fig. 5a, ii).

We also performed another control experiment with a matching piperidine concentration of 19±1 mM. In this case **8** is fully converted and **15** reaches a conversion of 20% (Fig. 5ab, ii).

Both the low and high concentration pairs of experiments are now reported in Fig. 5.

To determine what would happen if higher amplitudes of piperidine (up to 100 mM) could be obtained, we turned to simulations (Fig. S52). These indicate that in the absence of oscillations the slow reacting substrate is converted by 60%, while in an oscillating system only the fast-reacting substrate is converted.

*The levels of substrate conversion realized in our experiments (Table S13) confirm that oscillating catalysis allows catalytic processes to be more selective than catalysis without oscillations in the presence of multiple substrates. In both control experiments (Fig. 5b,i and 5b,ii) the ratio of conversion to products **10** and **16** is 5:1, which is increased by a factor 4 in the oscillation experiment with the lower piperidine concentration (15±3 mM, Fig. 5a,i) and even by a factor 10 with higher piperidine concentration (22±3 mM, Fig. 5a,ii). The improvements of selectivity in systems reaching larger catalyst amplitudes is expected to be even more pronounced (Fig. S52).*

These results are also discussed in the manuscript (highlighted in yellow) and the figure 5 (as well as SI) has been updated with new experiments accordingly.

Could the authors clarify what the two horizontal arrows next to the red curve (normalised piperidine absorbance) in both Figures 5a and 5b are indicating? Prior to the comments by Referee 3 about the comparison of normalised absorbance in Fig 5, it was assumed that these arrows indicate comparable concentrations of piperidine in each experiment (Fig 5a and 5b) but this is now potentially misleading. The caption of Fig 5 needs to be explicit as to the fact that the normalised concentrations of piperidine are not comparable between Fig 5a and 5b. Ideally this figure would include the authors estimates for the concentration of **1** in those experiments either in the caption text or the image (0-18 mM and 70 mM).

The red line is DBF (**6**), as in figure 4. This is the component of the core oscillator that we can still follow clearly in the catalytic oscillator. We have plotted it here to make it clear that in the case of Fig. 5a the system is oscillating and in the case of Fig. 5b it is not.

To make the figure 5 simpler, graphs corresponding to DBF formation have been removed and the caption text has been reduced. The detailed information of this figure is now in SI.

3. Quantification and side reactions

There seems to be some reasonable practical obstacles to monitoring the system, particularly in the flow apparatus and this is ascribed to overlapping signals (by NMR and/or IR). However, it isn't well explained why the method of NMR aliquots used for the batch experiments can't be used in the CSTR, can't the outflow material itself be used for analysis without disrupting the experiment?

While the same methods could in principle be used, we have refrained from doing so because taking samples large enough to yield reliable measurements from the flow reaction could disrupt the experiment, contrary to the batch experiment where the volume doesn't impact the course of the reaction.

Unfortunately we also cannot use the outflow material, due to the nature of our set-up. Since our oscillator releases CO₂, we cannot use a closed reactor which we can simply push liquid through. Instead, our set-up allows for the escape of gasses from the CSTR and extracts the liquid using a secondary syringe pump from which we cannot take samples.

The compound spectra provided in Figs S21-29 are appreciated, but it still isn't clear which signals are preventing the monitoring of the concentration of 16, do the authors mean when using NMR or IR?

In the IR the problem is the sheer number of carbonyls, and the low concentration of **16**, which prevents following any single one adequately. ¹H-NMR is discussed in detail later in the response to comments regarding NMR data.

The spectra in Figs S21-27 are in CDCl₃ rather than d₆-DMSO. It would be appropriate to provide spectra of these compounds in d₆-DMSO so that comparison of their NMR signals can be made to Fig S19 (and others). This is of particular importance for clarity on whether the signals of 16 and other species overlap in the reaction solvent.

Reference spectra of Piperidine (**1**), Fmoc-piperidine (**2**), *p*-nitrophenyl acetate (**3**), phenyl acetate (**4**), *N*-methylpiperidine (**5**), DBF (**6**), *N*-acetylpiperidine (**7**), dimethylmalonate (**8**), salicyl aldehyde (**9**), 3-(methoxycarbonyl)coumarin (**10**), *p*-nitrophenol (**12**), phenol (**14**), and *p*-hydroxybenzaldehyde (**15**) in DMSO-d₆ were added to the SI as Fig. S30-S42.

The offset presentation of the stacked NMR data in Figure S19 also does not make it obvious where the overlapping signals that obscure 16 could be. I suggest that offset and non-offset stacked NMR traces are both shown for the reaction timecourse so the obscured signals (or absence of signals from 16) can be better distinguished. According to its NMR spectrum (Fig S28), product 16 has signals at 6.84, 6.82, 3.80 and 3.75 ppm which are close to but sufficiently separated from other compounds' multiplets for some quantification. Can the authors explain whether any quantification of 16 is possible by NMR aliquots of the oscillating CSTR reaction (Figure 5a experiment), specifically using the signals of 16 described above?

The doublet at 3.80 and 3.75 gives only very weak signals in the oscillation experiment which are in close vicinity to the dimethylmalonate peak, its carbon satellites, and the peak of the methyl of 10. Because of this the peaks cannot be unambiguously assigned in the reaction spectra.

The doublet around 6.82 is also very weak in the oscillating experiment, although indeed separated sufficiently from the forest generated by all the aromatic rings. We have now used this signal (although it should be noted that the obtained integrals are rather noisy) to add traces of product **16** to Fig. 5.

In Figure S19 there is the growth of a sizeable multiplet at ~6.8-6.9 ppm but this doesn't obviously correlate to any of the characterised compounds in the mixture. Can the authors comment on what this compound is and whether it affects the system?

The multiplet corresponds to phenol, a product of the reaction between piperidine and phenyl acetate. A reference spectrum of phenol in DMSO- d_6 is included as Fig. S41. Moreover, the same peaks can be seen appearing in Fig. S1 (representative $^1\text{H-NMR}$ spectra of a pulse in batch), and Fig. S16 (representative $^1\text{H-NMR}$ spectra of a pulse in batch with Knoevenagel condensation). We have no reason to suspect that phenol affects the system as the much more acidic *p*-nitrophenol only has a very minor effect (Fig. S44B and Fig. S48).

Related, the authors explain in response to Referee 3 that the concentration of piperidine could not be determined in the experiments of Fig 5a due to error and dilution of the aliquots. Couldn't the concentration of piperidine and other components (e.g. 16) be readily determined by a more sensitive technique such as LCMS, HPLC or GC?

Although the referee's suggestion is an interesting idea it would require extensive calibration of at least all followed components and perhaps all components of the oscillator. This makes these methods experimentally challenging, with no guarantee of success. This is exacerbated by the fact that samples are quenched with TFA, which could prevent the piperidine from evaporating, ionizing, or eluting properly.

In all these cases we will still have the issue that the peak in piperidine is quite sharp and thus with samples we need to get very lucky to get a sample on or near the peak concentration. Determining the true range of these oscillations using a sample taking technique is thus quite tricky.

However, as mentioned earlier, we did attempt to derive the peak concentration of piperidine during the oscillation experiments in Fig. 5 (Fig.S54).

Have the authors also considered that 16 could be a potential acid-base inhibitor of piperidine (similar to *p*-nitrophenol inhibition which is acknowledged by the authors)? The hydroxyl of product 16 is expected to be more acidic than that of 15 and its pK_a probably resembles *p*-nitrophenol. This could be an underlying driver for the selection seen and/or why reaction with substrate 15 is so sensitive to catalyst concentration (e.g. large rate deceleration seen in Fig S40b when an equimolar amount of product is formed, relative to catalyst 1). This inhibition is not possible with coumarin product 10 and highlights why forming two different product classes could be problematic when benchmarking selectivity.

The pK_a of product **16** is predicted to be 9.3 ± 0.3 (<https://scifinder-n.cas.org/searchDetail/substance/63ea35e92e7dee60e347541e/substanceDetails>). This is, roughly, 2 above the pK_a of *p*-nitrophenol. Since we know that, although it does interact with piperidine, *p*-nitrophenol does not strongly affect either the oscillation itself (Fig. S44B) or the Knoevenagel condensation (Fig. S48) it is very unlikely that product **16** will.

a) Line 233 – Fig S32 should read Fig S34; b) Fig S38 – in the coloured caption above, both of the substrates are labelled Substrate 1; c) Fig S12 – caption says “CSTR pulse in batch” is this correct? Should this read ‘stirred pulse in batch’? rather than implying flow?

These mistakes have been corrected

Fig S38 – the use of both teal and green in the simulation plot makes matching the curves to the coloured compounds in the legend above a little tough, using an alternate colour would make understanding of this figure smoother.

We have changed the colors of this plot to red and blue instead.

Fig S40a – is the inclusion of *p*-nitrophenol in the control experiment for the Knoevenagel reaction between 8 and 9 to slow the kinetics for monitoring? Or does it have some other role besides being a weak acid?

We investigated the effect of the phenols on the Knoevenagel condensation at the suggestion of Referee #1 in the first round of revisions. As can be seen in Fig. S48 the presence of *p*-nitrophenol has very little influence on the rate of the reaction.

We investigated control experiments in the absence of *p*-nitrophenol but also in the presence of it as the latter is closer to the conditions of the oscillator. Since there is very little difference in the results we have changed the experiment included here from the one that contains *p*-nitrophenol to the one without.

Reviewer Reports on the Third Revision:

Referees' comments:

Referee #1 (Remarks to the Author):

In my view, the results in Fig5, supported by the discussion and simulations in the SI, convincingly show the claimed product selectivity when performing the reaction under oscillatory conditions. The authors have convincingly addressed the questions raised by the referees and have gone the extra mile to further support their findings, including extensive additional experiments. Overall, this is new and very clever development, pushing the boundaries of what can be done with chemical reaction networks. As such, I fully support publication.

Referee #2 (Remarks to the Author):

The authors seem reluctant to produce supported data - and have said they cannot carry out some experimental requests, but later agree its possible to do it after reviewers request again. For example, they couldn't evaluate concentration due to NMR complexity in the first round of revisions, but after pointing out a specific obvious NMR signal they can follow they do so in the second round of revision. Measuring the concentration of the active catalyst required two rounds of revisions and there is a similar reluctance in the refusal to use more sensitive techniques such as HPLC, saying they need to calibrate (straightforward for isolable compounds). There does not seem to be a good case against performing high frequency sampling for HPLC but it would bolster the strength of their claims.

The selectivity experiments remain misleading. In Figure 5 demonstrating selectivity at "high concentration" and "low concentration" of catalyst uses very similar concentrations (22 mM and 15mM respectively). These experiments were requested as controls as the original data from Figure 5 were not comparable due to large differences in concentration of the key catalyst between oscillating and non-oscillating experiments (~0-18 mM and 70 mM).

The authors state "Both the low and high concentration pairs of experiments are now reported in Fig. 5." Referring to a 15mM reaction as "low concentration" and a 22 mM reaction as "high concentration" is not appropriate. The authors highlight solubility problems (though they are ambiguous regarding what the specific issue is), all the same, their new experiments do not strengthen their case for selectivity. What about presenting graphs from either "low" concentration (i) or "high" concentration (ii) experiments, as the inclusion of all four graphs does not appear to add anything qualitatively and complicates the figure.

Editing of SI scheme? We (and Referee 1) previously raised queries relating to phenols and poisoning of the catalyst, which the authors have since stated have negligible effect on the system. Experiment in Fig S55a included p-nitrophenol as a poison to slow the rate of the reaction. Since we asked if product phenol 16 may have a similar effect, the authors have removed p-nitrophenol from the scheme of Fig S55a, stating that its impact was minimal, but the data shown is the identical to the

data when p-nitrophenol was present.

There is no doubt the system is interesting and the coupling of the oscillator to a secondary reaction is a valuable contribution, but there is insufficient evidence to support the claim that the selectivity of the coupled reaction should be attributed to the oscillation itself. Either the data supporting this claim needs to be experimentally meaningful, or the claim needs to be removed entirely. Although the complexity of this system makes it challenging to obtain clear evidence, it should be possible, and the evidence provided does not support the conclusion of 'oscillation-induced' selectivity.

Referee #3 (Remarks to the Author):

Since the last revision, the authors have added more information to corroborate the claim around selectivity through catalysis with the oscillatory system and they have quantified the concentrations of some species in Figure 5. I suppose the main issue is that it is still not completely clear that with the way the new system is currently set up, the desired outcome of selectivity can only be achieved under oscillatory conditions. Currently it seems that with the same maximum concentration of piperidine, the oscillatory case shows more selectivity (albeit by a small amount) than the steady state case. But in (b) in figure 5, if slightly less piperidine were employed under steady state conditions, would that give similar conversions to (a)? Or in Figure S52c – the simulations - less piperidine could be employed than currently shown on the plot to reach the point where the non-oscillatory case and oscillatory case coincide in terms of selectivity? Or am I missing something? Probably it would be better if chemoselectivity were only possible with oscillations, although the attempts to harness the system to perform useful function still represent an important direction in the development of these types of systems.

Suggest the authors rephrase - Pg 9: "in a conventional CSTR setup in which the catalyst is present continuously, the less reactive substrate can be freely converted after the more reactive substrate has reached full conversion". If the processes occur in parallel, then both will take place simultaneously. After an initial transient in a CSTR during which the more reactive species is consumed faster, a steady state should be achieved with relative amounts determined by the flow rate and kinetics. Full conversion could only be achieved in batch.

Author Rebuttals to Second Revision:

Referee #1 (Remarks to the Author):

In my view, the results in Fig5, supported by the discussion and simulations in the SI, convincingly show the claimed product selectivity when performing the reaction under oscillatory conditions. The authors have convincingly addressed the questions raised by the referees and have gone the extra mile to further support their findings, including extensive additional experiments. Overall, this is new and very clever development, pushing the boundaries of what can be done with chemical reaction networks. As such, I fully support publication.

We would like to thank the Referee for the highly positive evaluation and appreciation of the presented work as well as for very helpful recommendations that helped us to improve the manuscript.

Referee #2 (Remarks to the Author):

The authors seem reluctant to produce supported data - and have said they cannot carry out some experimental requests, but later agree it's possible to do it after reviewers request again.

We regret that the Referee devaluates our revision efforts, in striking contrast to Referee 1 who notes: *'The authors have gone the extra mile to further support their findings, including extensive additional experiments'*. We have endeavored to act in good faith throughout the reviewing process. In our responses to the referees' comments, we have carried out significant additional experimental work, provided detailed responses to the comments of referees and met the response deadlines set to us.

For example, they couldn't evaluate concentration due to NMR complexity in the first round of revisions, but after pointing out a specific obvious NMR signal they can follow they do so in the second round of revision. Measuring the concentration of the active catalyst required two rounds of revisions and there is a similar reluctance in the refusal to use more sensitive techniques such as HPLC, saying they need to calibrate (straightforward for isolable compounds). There does not seem to be a good case against performing high frequency sampling for HPLC but it would bolster the strength of their claims.

We once again thank the Referee for pointing out this NMR signal. We have since included it in all the figures showing oscillation enhanced selectivity.

We agree with the Referee that high frequency sampling and analysis with a secondary technique such as HPLC is likely possible and would serve as an additional tool to support our claims. However, the provided data, using both in-situ react-IR as well as ^1H -NMR spectroscopy, also sufficiently support what is claimed in the manuscript. We were hesitant to commit to a completely different, additional method of sampling and following the system in time because the two different methods used already support our findings/claims and we needed to adhere to the revision timeframe.

The selectivity experiments remain misleading. In Figure 5 demonstrating selectivity at "high concentration" and "low concentration" of catalyst uses very similar concentrations (22 mM and 15mM respectively). These experiments were requested as controls as the original data from Figure 5 were not comparable due to large differences in concentration of the key catalyst between oscillating and non-oscillating experiments (~0-18 mM and 70 mM).

The authors state "Both the low and high concentration pairs of experiments are now reported in Fig. 5." Referring to a 15mM reaction as "low concentration" and a 22 mM reaction as "high concentration" is not appropriate.

Whether something can be called high or low is subjective and depends on the system under study. It has proven impossible to increase the 'high concentration' experiment beyond 22mM without affecting the dynamics of the system. Increasing the concentration of Fmoc piperidine requires an increase of the concentration of the slow inhibitor as well, which would turn it, apart from being another reagent, into a cosolvent in this system, due to its increased volume. This would significantly change the polarity of the whole system, altering the rates of all the reactions involved and consequently the oscillation space.

The authors highlight solubility problems (though they are ambiguous regarding what the specific issue is), all the same, their new experiments do not strengthen their case for selectivity.

For the flow experiments we use a two-syringe setup, where the flow rate of the syringes is the same. The consequence of this is that the concentration of the components of the oscillator in the syringe is twice that in the reactor. As a result, we run into a solubility limit sooner than would be expected from the concentrations in the reactor. Main culprit in this regard is Fmoc-piperidine. We were able to dissolve it up to 0.4 M, but for 70 mM amplitude of piperidine oscillations we would need 0.8 M. Our attempts to dissolve the components at this concentration were unsuccessful.

What about presenting graphs from either “low” concentration (i) or “high” concentration (ii) experiments, as the inclusion of all four graphs does not appear to add anything qualitatively and complicates the figure.

We had included both figures in the main text to show how the selectivity enhancement increases with increasing amplitude. As per the Referee’s suggestion we have moved the lower concentration example to the SI (Fig. S36). Additionally, we have focused the discussion in the main text on the higher concentration example thereby clarifying the main point.

Editing of SI scheme? We (and Referee 1) previously raised queries relating to phenols and poisoning of the catalyst, which the authors have since stated have negligible effect on the system. Experiment in Fig S55a included p-nitrophenol as a poison to slow the rate of the reaction. Since we asked if product phenol **16** may have a similar effect, the authors have removed p-nitrophenol from the scheme of Fig S55a, stating that its impact was minimal, but the data shown is the identical to the data when p-nitrophenol was present.

Referee 1 asked us to investigate the influence of phenols on the Knoevenagel condensation in the first round of revisions. Ever since there has been a figure in the SI (currently Fig. S23, but Fig. S35 and Fig. S48 in preceding 2nd and 3rd revisions) that shows that phenol has no influence and *p*-nitrophenol has a mild inhibitory effect, not a poison. During the 2nd revision Fig 38a (Fig. S40a and Fig. S55a in preceding 2nd and 3rd revisions) was presented erroneously with p-nitrophenol indicated under the reaction arrow, although the reaction was done without it. Therefore, in the revised SI (3rd and current revision) the mistake has been corrected by removing p-nitrophenol from the scheme.

The influence of the electron-withdrawing nitro group directly adjacent to the aromatic ring should make p-nitrophenol significantly more acidic than Knoevenagel product **16**, where the electron-withdrawing groups are separated from the aromatic ring by two bonds (a C-C and a C=C bond). Therefore, the inhibitory effect of **16** on the catalysis will be much smaller than that of p-nitrophenol.

There is no doubt the system is interesting and the coupling of the oscillator to a secondary reaction is a valuable contribution, but there is insufficient evidence to support the claim that the selectivity of the coupled reaction should be attributed to the oscillation itself. Either the data supporting this claim needs to be experimentally meaningful, or the claim needs to be removed entirely. Although the complexity of this system makes it challenging to obtain clear evidence, it should be possible, and the evidence provided does not support the conclusion of ‘oscillation-induced’ selectivity.

We have rephrased the claim to *oscillation enhanced selectivity* to more accurately describe the observed phenomenon.

Referee #3 (Remarks to the Author):

Since the last revision, the authors have added more information to corroborate the claim around selectivity through catalysis with the oscillatory system and they have quantified the concentrations of some species in Figure 5. I suppose the main issue is that it is still not completely clear that with the way the new system is currently set up, the desired outcome of selectivity can only be achieved under oscillatory conditions. Currently it seems that with the same maximum concentration of piperidine, the oscillatory case shows more selectivity (albeit by a small amount) than the steady state case. But in (b) in figure 5, if slightly less piperidine were employed under steady state conditions, would that give similar conversions to (a)? Or in Figure S52c – the simulations - less piperidine could be employed than currently shown on the plot to reach the point where the non-oscillatory case and oscillatory case coincide in terms of selectivity? Or am I missing something? Probably it would be better if chemoselectivity were only possible with oscillations, although the attempts to harness the system to perform useful function still represent an important direction in the development of these types of systems.

We would like to thank the Referee for recognizing the importance of the work and for all the useful suggestions. Following the Referee's line of thought, we did additional experiments to determine whether there is a point where the non-oscillatory case and oscillatory case coincide in terms of selectivity. We carried out control experiments with lower steady state [1] (at 5 mM and 1mM in the absence of oscillator components, Fig. S37) and found that it is indeed possible to obtain similar selectivity with traditional optimization. However, if the reaction is part of a more extensive chemical reaction network such optimization may not be possible. That with oscillations the desired selectivity can be obtained for a much larger range of catalyst concentrations is an attractive feature in these cases. To describe the presented selectivity phenomenon more accurately we have replaced the phrase oscillation *induced* selectivity by oscillation *enhanced* selectivity.

Suggest the authors rephrase - Pg 9: "in a conventional CSTR setup in which the catalyst is present continuously, the less reactive substrate can be freely converted after the more reactive substrate has reached full conversion". If the processes occur in parallel, then both will take place simultaneously. After an initial transient in a CSTR during which the more reactive species is consumed faster, a steady state should be achieved with relative amounts determined by the flow rate and kinetics. Full conversion could only be achieved in batch.

We have rephrased the sentence according to the Referee's suggestions.